



# The International Soil Moisture Network: serving Earth system science for over a decade

Wouter Dorigo[1], Irene Himmelbauer[1], Daniel Aberer[1], Lukas Schremmer[1], Ivana Petrakovic[1], Luca Zappa[1], Wolfgang Preimesberger[1], Angelika Xaver[1], Frank Annor[2,3], Jonas Ardö[4], Dennis Baldocchi[5], Günter Blöschl[6], Heye Bogena[7], Luca Brocca[8], Jean-Christophe Calvet[9], Julio J. Camarero[10], Giorgio Capello[11], Minha Choi[12], Michael C. Cosh[13], Jerome Demarty[14], Nick van de Giesen[3], Istvan Hajdu[15], Karsten H. Jensen[16], Kasturi Devi Kanniah[17], Ileen de Kat[18], Gottfried Kirchengast[19], Pankaj Kumar Rai[20], Jenni Kyrouac[21], Kristine Larson[22], Suxia Liu[23], Alexander Loew[24,†], Mahta Moghaddam[25], José Martínez Fernández[26], Cristian Mattar Bader[27], Renato Morbidelli[28], Jan P. Musial[29], Elise Osenga[30], Michael A. Palecki[31], Isabella Pfeil[1], Jarrett Powers[32], Jaakko Ikonen[33], Alan Robock[34], Christoph Rüdiger[35], Udo Rummel[36], Michael Strobel[37], Zhongbo Su[38], Ryan Sullivan[21], Torbern Tagesson[4,16], Mariette Vreugdenhil[1], Jeffrey Walker[35], Jean Pierre Wigneron[39], Mel Woods[40], Kun Yang[41], Xiang Zhang[42], Marek Zreda[43], Stephan Dietrich[44], Alexander Gruber[45], Peter van Oevelen[46], Wolfgang Wagner[1], Klaus Scipal[47], Matthias Drusch[48], and Roberto Sabia[47]

[1]Department of Geodesy and Geoinformation, TU Wien, Vienna, Austria
[2]Trans-African Hydro-Meteorological Observatory, Delft, The Netherlands
[3]Department of Water Resources, Delft University of Technology, Delft, Netherlands
[4]Department of Physical Geography and Ecosystem Science, Lund University, Lund, Sweden
[5]Department of Environmental Science, Policy and Management, University of California, Berkeley, CA, United States
[6]Institute of Hydraulic Engineering and Water Resources Management, TU Wien, Vienna, Austria
[7]Forschungszentrum Juelich GmbH, Juelich, Germany
[8]Research Institute for Geo-Hydrological Protection, CNR, Perugia, Italy
[9]CNRM, Université de Toulouse, Météo-France, CNRS, Toulouse, France
[10]Instituto Pirenaico de Ecología, IPE-CSIC, Zaragoza, Spain
[11]Institute of Sciences and Technologies for Sustainable Energy and Mobility, National Research Council of Italy, Torino, Italy
[12]Department of Water Resources, Graduate School of Water Resources, Sungkyunkwan University, Suwon, Republic of Korea
[13]USDA-Agricultural Research Service, Hydrology and Remote Sensing Laboratory, Beltsville, MD, United States
[14]Hydrosciences Montpellier, IRD, CNRS, Universite Montpellier, France
[15]PlantTech Research Institute, Tauranga, New Zealand
[16]Department of Geosciences and Natural Resource Management, University of Copenhagen, Denmark
[17]Research Institute for Sustainable Environment, Universiti Teknologi Malaysia, Johor Bahru, Malaysia
[18]VanderSat B.V., Haarlem, The Netherlands
[19]Wegener Center for Climate and Global Change and Institute of Physics, University of Graz, Austria
[20]Indian Institute of Technology Kanpur, India
[21]Environmental Science Division, Argonne National Laboratory, Lemont, IL, United States
[22]University of Colorado, Boulder, CO, United States
[23]Institute of Geographic Sciences and Natural Resources Research, Chinese Academy of Sciences, Beijing, China; College of Resources and Environment/Sino-Danish Center, University of Chinese Academy of Sciences, Beijing, China
[24]Department of Geography, Ludwig-Maximilian-Universität, Munich, Germany





[25]Ming Hsieh Department of Electrical and Computer Engineering, University of Southern California, Los Angeles, CA, United States

[26]Instituto Hispano Luso de Investigaciones Agrarias, CIALE, Universidad de Salamanca, Villamayor, Spain

[27]Laboratory of Analysis of the Biosphere, University of Chile, Santiago, Chile

[28]Department of Civil and Environmental Engineering, Perugia University, Italy

[29]Institute of Geodesy and Cartography, Warsaw, Poland

[30]Aspen Global Change Institute, Basalt, CO, United States

[31]NOAA National Centers for Environmental Information, Asheville, NC, United States

[32]Agriculture and Agri-Food Canada - Science and Technology Branch, Winnipeg, MB, Canada

[33]Finnish Meteorological Institute, Space and Earth Observation Centre, Helsinki, Finland

[34]Department of Environmental Sciences, Rutgers University, New Brunswick, NJ, United States

[35]Department of Civil Engineering, Monash University, Clayton, Australia

[36]DWD, Meteorologisches Observatorium Lindenberg – Richard-Aßmann-Observatorium, Lindenberg, Germany

[37]USDA-NRCS National Water and Climate Center, Portland, OR, United States

[38]University of Twente, Faculty of Geo-Information Science and Earth Observation, Enschede, the Netherlands

[39]MOST, ISPA, INRAE Bordeaux Aquitaine, France

[40]University of Dundee, UK

[41]Department of Earth System Science, Tsinghua University, Beijing, China

[42]State Key Laboratory of Information Engineering in Surveying, Wuhan University, Wuhan, China

[43]Department of Hydrology and Atmospheric Sciences, University of Arizona, Tucson, AZ, United States

[44]Federal Institute of Hydrology, Koblenz, Germany

[45]Department of Earth and Environmental Sciences, KU Leuven, Heverlee, Belgium

[46]International GEWEX Project Office, Washington, DC, United States

[47]European Space Agency, ESRIN, Frascati, Italy

[48]European Space Agency, ESTEC, Noordwijk, The Netherlands

[†]Deceased July 02, 2017

**Correspondence:** Wouter Dorigo (wouter.dorigo@tuwien.ac.at)

**Abstract.**

In 2009, the International Soil Moisture Network (ISMN) was initiated as a community effort, funded by the European Space Agency, to serve as a centralised data hosting facility for globally available in situ soil moisture measurements (Dorigo et al., 2011b, a). The ISMN brings together in situ soil moisture measurements collected and freely shared by a multitude of

organisations, harmonizes them in terms of units and sampling rates, applies advanced quality control, and stores them in a database. Users can freely retrieve the data from this database through an online web portal (ismn.earth). Meanwhile, the ISMN has evolved into the primary in situ soil moisture reference database worldwide, as evidenced by more than 3000 active users and over 1000 scientific publications referencing the data sets provided by the network. As of December 2020, the ISMN now contains data of 65 networks and 2678 stations located all over the globe, with a time period spanning from 1952 to present.

The number of networks and stations covered by the ISMN is still growing and many of the data sets contained in the database continue to be updated. The main scope of this paper is to inform readers about the evolution of the ISMN over the past decade, including a description of network and data set updates and quality control procedures. A comprehensive review of existing literature making use of ISMN data is also provided in order to identify current limitations in functionality and data usage, and to shape priorities for the next decade of operations of this unique community-based data repository.



# 1 Introduction

Ground-based soil moisture measurements of land surface variables are indispensable in the process of developing, validating, and advancing spatially contiguous data sets derived from satellites or models (Loew et al., 2017; Gruber et al., 2020). Although the first systematic measurements of soil moisture started well before the satellite era in the former Soviet Union to support agricultural decision making (Robock et al., 2000), it was not until the early 2000s that soil moisture monitoring networks

started being widely established as part of hydrological and meteorological observing capacities. Particularly, the launch of the Soil Moisture Ocean Salinity (SMOS) mission of the European Space Agency (ESA) in 2009 (Kerr et al., 2016), and the launch of the Soil Moisture Active Passive (SMAP) mission of the National Aeronautics and Space Administration (NASA) in 2015 (Entekhabi et al., 2010), boosted the establishment of new research networks (Colliander et al., 2017).

While all networks are a valuable asset for assessing the skill of soil moisture products under various conditions and scales,

their usage is hampered by the diversity of sensors, data formats, quality control, and accessibility mechanisms. The need of bringing together and harmonising available soil moisture data was recognised by the international soil moisture community and expedited by the Global Energy and Water cycle Exchanges (GEWEX) project of the World Climate Research Programme (WCRP) with support of the Commission on Earth Observation Satellites (CEOS), the Global Climate Observing System (GCOS), and the Group on Earth Observation (GEO). In the advent of the SMOS mission, ESA decided to provide the financial

impetus to establish a global reference database of in situ soil moisture measurements for the purpose of satellite product development and validation. As a result, the International Soil Moisture Network (ISMN) went online in 2010 (Dorigo et al., 2011b, a).

The primary objective of the ISMN is to collect in situ soil moisture data sets shared by various data organisations on a voluntary basis and make them available in a harmonized format through a centralised free and open web portal (ismn.earth).

While 10 years after its launch the core objective of the ISMN remains valid, its functionality has expanded since then. This new functionality includes the integration of advanced quality control methods (Dorigo et al., 2013), the provision of additional metadata and ancillary variables (e.g., precipitation, soil and air temperature), ongoing automation, the provision of software code to users, and the implementation of various tools to promote the information exchange between users, the ISMN, and the data providers. Moreover, the ISMN has substantially grown in terms of networks, stations, and data sets.

Data from the ISMN has supported hundreds of scientific papers on soil moisture, satellite product and land surface model validation in particular (e.g., Al-Yaari et al. (2019b); Brocca et al. (2014a); Beck et al. (2020)). Several operational data producing services routinely access the ISMN data for repeated quality assurance, including ESA's Climate Change Initiative (Dorigo et al., 2017), the Copernicus Global Land Service (Bauer-Marschallinger et al., 2018), and the Copernicus Climate Change Service (C3S; Dorigo et al. (2017)). Other domains have also exploited the ISMN data, e.g. in meteorology, drought

monitoring, or land-atmosphere coupling (Section 4).

Despite the valuable contribution of the ISMN to satellite and climate communities, multiple challenges have yet to be mastered, including the heterogeneous availability in space and time (Dorigo et al., 2015), scale differences between in situ measurements and satellite or model sampling (Gruber et al., 2013), full characterisation and traceability of uncertainties,



and differences in spatial-temporal support of the observations caused by different measurement techniques and landscape

heterogeneity (Ochsner et al., 2013). New scientific avenues to improve the spatial coverage could be the inclusion of soil moisture data sets from low cost sensors collected by citizens. For climate applications, stable long-term reference data are required, calling for the coordinated establishment and maintenance of Fiducial Reference Measurement (FRM) stations, as outlined by the GEO/CEOS Quality Assurance framework for Earth Observation (QA4EO) (GCOS, 2016; Montzka et al., 2020; Gruber et al., 2020).

The scope of this paper is to inform readers about the evolution of the ISMN over the past decade, including a description of network and data set updates, new quality control procedures, and new functionality of the data portal. We also review scientific literature making use of ISMN data to assess the achievements facilitated by the ISMN and to identify current limitations in data availability and functionality, and challenges in data provision and use. Based on this review, prerequisites and priorities needed to ensure another decade of this unique community-based data repository are defined.

## 60   2   The ISMN data hosting facility

Although the ISMN may be considered a mere data repository, there is much more to it: its core functionality includes collecting data from participating data providing networks, harmonizing them in terms of units, sampling rates, naming, and metadata, performing automated quality control, storing the data and metadata in a queryable database, and making them available through a web interface. And from a system perspective, it entails even more, e.g., communication with (potential)

data providers and users (Appendix C1).

### 2.1   Data and metadata summary

Currently, the ISMN contains 65 networks providing access to a total of 2678 stations, approximately 10,000 soil moisture data sets, and an additional 10,000 data sets of other meteorological variables (collocated with the soil moisture measurements). Although the ISMN is a global network, most networks and stations are located in North America, Europe, Australia and Asia

(Figure 1). New networks continuously being added, while many existing networks are still being upgraded with additional stations, soil moisture sensors, and meteorological variables. The diversity of networks is large, ranging from networks with a single station to networks comprising more than 400 stations covering different landscape types as well as periods.

Most of the networks originate from scientific initiatives in various disciplines (e.g., remote sensing, soil sciences, agriculture, meteorology), and only a few are run by operational entities like national weather or environmental services. Consequently,

a lack of sustainable project funding has forced several scientific networks to close after some time. 17 out of the 65 networks contained in the ISMN have become inactive and will no longer provide data set updates (Figure 2). Data go back as far as 1952 but none of the data sets spans the entire 70 year period. The longest available time series ( 40 years) is provided by the RUSWET-AGRO network in the former Soviet Union, while the longest operating network still active is SNOTEL in the United States.



**Figure 1.** Locations of ISMN networks and sites plotted with the ISMN package described in the *Code and data availability* section [Status December 2020].





Most networks provide data set updates at yearly or irregular intervals. data sets from six networks (COSMOS, FMI, SCAN, SNOTEL, USCRN, WEGENERNET), comprising approximately 900 stations, are updated in near-real-time (NRT; status December 2020), which is currently defined as once per day. While the earliest networks were sampled manually at weekly, fortnightly, or even monthly intervals, most current networks take their measurements using electronic devices at daily, hourly or even more frequent sampling rates. For more details on the networks see Appendix A, Dorigo et al. (2011b), Dorigo et al.

(2013) or the individual references herein.

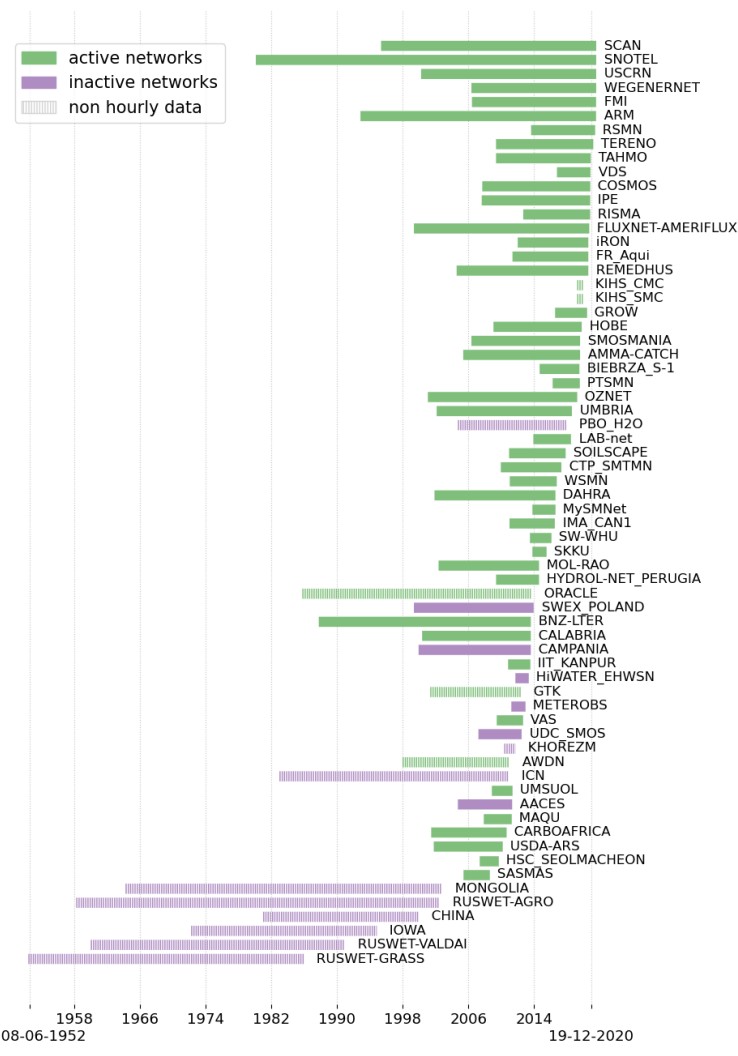

**Figure 2.** Overview of all available networks, the individual time span of the data availability within the ISMN, and their operational status (Status December 2020).



The variables contained in the ISMN (Table 1 and 2) originate from networks that were built for various purposes, which consequently do not all contribute the same information. Since the ISMN was initially founded as a validation database for satellite (surface) soil moisture data, each station in the database provides soil moisture for the upper soil layer ($\leq 0.10$ meter depth). Soil moisture data sets in the ISMN can go as deep as two metres, but generally with a decreasing number of measure-
ments locations with depth (Table D1). Some stations deploy more than one sensor at a certain depth, either as replacement in case of failure of one of the sensors or to characterise local soil moisture variability. The availability of meteorological data like precipitation, soil and air temperature, is even more heterogeneous, depending on the scope of the network or the data sharing policy of the data providing organisation. It is also quite common that single time series in the database are composed of the consecutive measurements of two or more different sensors, when a sensor is replaced after failure.

**Table 1.** Overview of all available temporally dynamic variables stored in the ISMN database. *Note that for precipitation and air temperature the measurement height above the ground surface is indicated.

| variable name | abbreviation | units | measurement depth* [m] | variable with depth? |
|---|---|---|---|---|
| soil moisture | sm | $\mathrm{m^3 m^{-3}}$ | 0.00 - 2.10 | Y |
| soil suction | su | kPa | 0.04 - 0.75 | Y |
| soil temperature | ts | °C | 0.00 - 2.03 | Y |
| air temperature | ta | °C | 2.00 - 12.00 | Y |
| surface temperature | tsf | °C | 0.00 - 0.00 | N |
| precipitation | p | mm | 0.00 - 2.00 | N |
| snow depth | sd | mm | 0.00 | N |
| snow water equivalent | sweq | mm | 0.00 | N |

**Table 2.** Overview of all available temporally static variables stored in the ISMN database.

| variable name | abbreviation | units | measurement depth [m] | sensors dependency |
|---|---|---|---|---|
| climate classification | clcl | none | none | N |
| land cover classification | lccl | none | on surface and above | N |
| soil classification | socl | none | none | N |
| bulk density | bd | $\mathrm{g\,cm^{-3}}$ | 0.00 - 1.50 | Y |
| sand fraction | sa | % weight | 0.00 - 1.50 | Y |
| silt fraction | si | % weight | 0.00 - 1.50 | Y |
| clay fraction | cl | % weight | 0.00 - 1.50 | Y |
| organic carbon | oc | % weight | 0.00 - 1.00 | Y |
| saturation | sat | % volume | 0.00 - 1.00 | Y |

*Table – continued on next page*





**Table 2.** Overview of all available temporally static variables stored in the ISMN database.

| variable name | abbreviation | units | measurement depth [m] | sensors dependency |
|---|---|---|---|---|
| field capacity | fc | % volume | 0.00 - 2.00 | Y |
| potential plant available water | ppaw | % volume | 0.00 - 2.00 | Y |
| permanent wilting point | wp | % volume | 0.00 - 2.00 | Y |

Metadata information can be divided into two categories, i.e., mandatory metadata, which allow for an unambiguous identifi-
cation of each network, station, and measurement in the ISMN database (Figure E1), and optional metadata, shared by data
providers to allow more in-depth analysis of their data sets. To be consistent between sites, the mandatory variables climate,
land cover, and soil characteristics are taken from external databases: Climate classification is taken from the Köppen-Geiger
database with a resolution of 0.1° (Peel et al., 2007); Dynamically evolving land cover for 2000, 2005, and 2010 is obtained
from ESA's Climate Change Initiative (CCI) land cover v1.6.1 with 300 meter resolution; Soil information is retrieved from
the Harmonized World Soil Database (HWSD v1.1; FAO/IIASA/ISRIC/ISS-CAS/JRC, 2009) with a 30" sampling, although
the actual resolution may strongly vary from location to location.

   If available, data providers can optionally share their own, more detailed, characterisations of land cover, soil, and qual-
ity flags with the ISMN. These are stored in addition to the same variables from external sources. All static variables per
measurement site and depth are listed in Table 2.

## 2.2   Data collection and harmonisation

Data collection is done either manually (mostly by email) or automated, depending on the degree of automation at the data
provider side. Although standards for in situ soil moisture data collection are available (Colliander et al., 2017; Montzka et al.,
2020), there is no general agreement within the community, neither is it prescriptive for participation in the ISMN. Thus, the
data being contributed to the ISMN are heterogeneous with regard to units (e.g., volumetric soil moisture, water depth, mass,
soil saturation, plant available water), installation depth, integration length, and positioning of the sensors (vertical, horizontal),
the metrical system, the sampling interval, and the time zones used for the measurement time stamps.

   The first harmonisation step for all data and metadata involves the conversion of units to internationally agreed scientific
units (e.g., m and °C). Then, following the recommendations of the World Meteorological Organization for weather observation
and forecasting (Williams, 2010), all data are resampled to hourly UTC reference time steps. Data that are available at higher
sampling rates are thinned by selecting the individual measurements at the hourly UTC reference time step or the ones that
are closest in time within a window of +/- 0.5 h. If there is no measurement available within this interval, the respective time
step is not stored in the database but can be recreated and filled with a no-data value upon download. The temporal resampling
scheme is valid for all included dynamic variables, except precipitation. Since precipitation is a flux and not a state variable,
all measurements within the hourly interval are summed to represent the full amount of precipitation within the preceding hour
(Dorigo et al., 2011b).


All soil moisture measurements provided to the ISMN are converted to volumetric soil moisture in $m^3 m^{-3}$. Since the majority of networks already shares their measurements in volumetric soil moisture units often no unit conversion is needed. For the other (mostly historical) networks, measurements are converted to volumetric units using metadata on soil properties (in

case measurements are provided as saturation level or plant available water) and/or the thickness of the soil layer represented by the measurement (in case measurements are provided as water height or mass) (Dorigo et al., 2011b).

The harmonisation of measurement depths is particularly challenging, as different networks adopt different measurement depths, similar sensors are positioned differently (horizontally vs. vertically), or their measurements represent different observation volumes, which may even differ according to the soil wetness (as for cosmic ray probes). Thus, harmonising soil

moisture measurements to one or several reference depths would require either assumptions on the measurements and soil properties or supplemental soil modelling. Additionally, since there are lots of potential uses for the data, there is no common agreement on the optimum sampling depths. For example, satellite calibration-validation generally requires observations of the 0-5 cm layer while land surface model evaluation requires reference measurements that are representative for the layers defined in the model (Dorigo et al., 2011b). Consequently, the ISMN does not harmonise measurement depths.

After data harmonisation, the data sets are submitted to extensive quality control procedures (Section 3. After quality control, all data sets of soil moisture and other variables, metadata information on networks, responsible organisations, sites, sensors as and static soil attributes for each station are stored in a relational database.

## 2.3    Data portal

The ISMN can be accessed at ismn.earth and consists of a project website containing, e.g., information about networks, data,

quality control, and partners, and a data interface where users can view, query, and download the data contained in the database.

The data interface displays the location and information of networks and single stations, and allows for plotting the available data to gain an impression of data availability and quality. Data can be selected for time period, area, single networks, or entire continents. Alternatively, the data download can be selected via an advanced SQL query, which allows users to make more specific selections (e.g., for sensor brand, or a certain depth range).

The selected data are directly extracted from the database and downloads are organised per network. For each network, the download contains (i) the measurements and their quality flags, (ii) information about the file data organisation, (iii) a description of the ISMN quality flags, (iv) a metadata file compliant with ISO 19115 and INSPIRE (Infrastructure for Spatial Information in the European Community) metadata standards, and (v) information about static site characteristics (e.g., land cover, climate class, soil characteristics).

The data set files are formatted according to the CEOP (Coordinated Energy and water cycle Observations Project) (Williams, 2010) standard, or a slightly modified version of it, in order to improve data usability (Dorigo et al., 2011b). These standards use the ascii format but a NetCDF format is foreseen for the near future.



## 2.4 Data provider and user involvement

The ISMN is entirely built on the voluntary, free-of-charge contributions from scientific and operational providers. This pre-
vents the ISMN from being too prescriptive towards the data providers in terms of delivery intervals, automation and formatting.
Hence, a careful balance is needed between inclusiveness on the one hand and data quality standards on the other. The ISMN
facilitates between users and data providers by reporting data quality issues and user feedback to the providers very six months.
This is done by means of a report on data usage statistics for each individual network, e.g., on the number of downloads, the
usage of their data in scientific publications, and flagging statistics. Together with obtaining visibility and citations, obtaining
feedback on data usage and data quality is one of the primary motivations for data providers to join the ISMN.

More than 3200 active users have registered since 2009 (status December 2020). Data download is free of charge but
user registration (compliant with the latest GDPR privacy standards) is required to prevent misuse of the data and to track
(undisclosed) statistics on data usage for the half-yearly provider reports.

News feeds on the ISMN web page and a bi-annual newsletter informs the users about new networks, new data sets, data
quality issues, important publications, workshops, and more. In addition, a dedicated forum and classical email exchange allow
users to raise and get response to issues. Moreover, open source software packages are available for reading and plotting the
data (Section 6).

## 3 Quality control

### 3.1 Quality flagging methodology

The wide variety of sensors types and installations, measurement protocols, calibration methods, preprocessing, and quality
control procedures adopted by the data providers result in data sets with large differences in quality and filtering. In an attempt to
harmonise the reliability of the data from different networks and sensors and to allow for marking spurious observations in near-
real-time, the ISMN has adopted automated quality procedures, which are applied to all observations feeding into the ISMN
(Dorigo et al., 2013). It uses several approaches to detect dubious soil moisture measurements, which can be subdivided into
geophysical dynamic range verification, geophysical consistency methods, and spectrum-based methods (Table 3). While the
first category of methods applies simple threshold checks directly to the measurements, the geophysical consistency methods
make use of ancillary data, either observed in situ at the same site or derived from model data (i.e., GLDAS-Noah). The
spectrum-based flags are the result of a series of conditions applied to the soil moisture measurement time series and their
first and second derivatives. The geophysical consistency and spectrum-based methods are only applied to the soil moisture
observations, while geophysical dynamic range verification is applied to all dynamic variables in the database.

Recently, a few refinements of the original flagging procedures as described and assessed in Dorigo et al. (2013) were
implemented to increase the flagging accuracy:

– The outlier detection method (Flag D06) now allows spikes to last two consecutive time steps instead of the initial one
 hour. This occurs when all conditions of Eq.[4-6] in Dorigo et al. (2013) are met, but the peak value remains unchanged

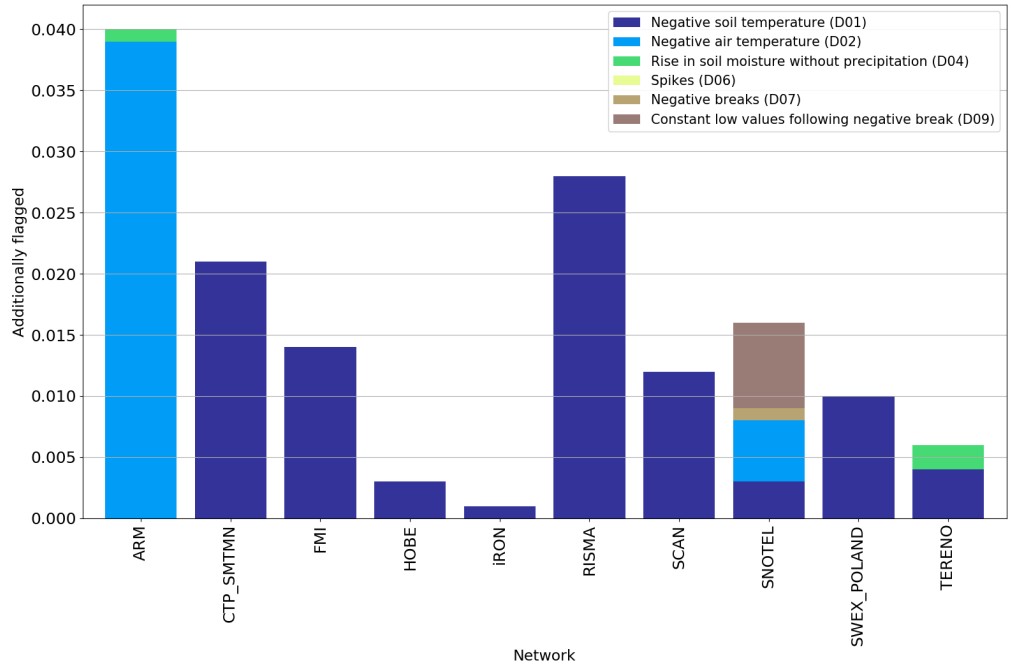

**Figure 3.** Additionally flagged observations by network shown as fraction of total number of observations. Note that, in principle, the cumulative sum of fractions can be >1 as the occurrence of each flag ranges between 0 and 1.

for an additional hour. The overall impact is small (flagging numbers increase from 0.30% to 0.33%) but its impact on extreme values can be significant.

– Flag D07 ("negative breaks (drops)") was extended with an extra possibility, which detects drops from values greater than 0.05 to exactly 0 $m^3 m^{-3}$ soil moisture:

$$x_t \geq 0.05 \land x_{t+1} = 0 \tag{1}$$

Since a spurious soil moisture drop is precondition for a low plateau (D09; "constant low values following negative break"), also the latter is affected. Adding the extra drop detection increased flagging numbers for D07 from 0.01 to 0.03% and for D09 from 0.9 to 1.1%.

– In case more than one soil temperature, air temperature, or precipitation sensor is available at a site, a flag is raised when at least one of them indicates a spurious observation, which leads to a small overall increase in flags D01, D02 and D04
(< 0.7%).





All quality control procedures adopted by the ISMN have been made available under the open source license agreement (see Section 6, https://github.com/TUW-GEO/flagit).

We assessed the refined flagging procedures by applying them to 10 networks with hourly data that include stations with multiple soil temperature, air temperature, or precipitation sensors. Despite the very low overall impact of the refined flags, for some networks they are substantial (Figure 3).

Measurements that are detected as erroneous by the quality control procedures are not deleted from the database, but tagged as such (Table 3). The flag is provided as additional attribute ("ISMN Quality Flag") to the observation upon download and can take one of the main categories C (exceeding plausible geophysical range), D (questionable/dubious), M (missing) or G (good). The D flag is raised when either a geophysical-consistency or a spectrum-based check is positive. An additional number indicates the actual cause for flagging (Table 3). A soil moisture observation may receive multiple C- and D-type flags but the good and missing flags are exclusive. Seven networks provide their own soil moisture quality flags, which are added to the ISMN database in addition to the ISMN flags common to all time series. Examples are flags for data quality (without further methodological description) or simple thresholds. For instance, RISMA tags soil moisture observations for frozen soils when the average temperature of two adjacent soil layers is below 0°C (Pacheco et al., 2014). By contrast, the ISMN flag D01 "Soil temperature <0°C" only considers the corresponding depth. For RISMA, the frozen soil flags provided by the network and those computed by the ISMN (D01 "soil temperature <0°C") agree for 87.8%.

## 3.2 Flagging occurrence

The most commonly raised flag is when one of the ancillary temperature observations, i.e., in situ soil temperature (D01), in situ air temperature (D02), or GLDAS soil temperature (D03) is <0°C (Table 3; Figure 4). Since in situ temperature measurements are not available for all networks and to keep consistency between networks, flags D01 and D02 are not shown in Figure 4. The number of observations flagged as frozen soil are not an indicator of the site in general but show which networks are located in areas with a pronounced cold season, e,g., stations from the FMI, RISMA, MAQU, SCAN, and SNOTEL networks.

The second-most common flag is C03 " soil moisture above the site-specific saturation point", which is computed from the HWSD soil properties. The site-specific saturation point is usually lower than the globally adopted less conservative threshold of 0.6 m³/m³ (flag C02) and thus raised more often (Figure 4). However, the HWSD soil properties are uncertain and consequently the C03 flag should be considered as indicative rather than as absolute qulaity indicator. Values exceeding the saturation point are often an indication of calibration biases or atypical site conditions. For example, the large number of C03 flags obtained for BIEBRZA-S-1 network is because it is installed in a temporarily flooded marshlands with peat porosity exceeding 80%.

Constant values as a result of saturation plateaus (D10) or after a negative break (D09) are the most common spectrum-based flags (Figure 5). The latter are often a sign of sensor drop outs and mostly limited to the GROW, ARM, HYDROL-NET-PERUGIA, and USDA-ARS networks. For example, Xaver et al. (2020) evaluated the sensors used in the GROW network and found occasional drops in soil moisture to zero, which may be the result of corroding contacts. Spikes and breaks are by nature





**Table 3.** Occurrence of flags for all variables and measurements contained in the ISMN (Status December 2020). The soil moisture flags are not exclusive, i.e., an observation can be tagged with multiple flags.

| Variable | Flag | Type | Description | Occurrence [%] |
|---|---|---|---|---|
| Soil moisture | C01 | range verification | soil moisture $< 0$ m$^3$/m$^3$ | 0.05 |
| | C02 | range verification | soil moisture $> 0.60$ m$^3$/m$^3$ | 1.17 |
| | C03 | range verification | soil moisture $>$ saturation point (based on HWSD) | 3.99 |
| | D01 | geophysical consistency | negative soil temperature (in situ) | 6.39 |
| | D02 | geophysical consistency | negative air temperature (in situ) | 16.57 |
| | D03 | geophysical consistency | negative soil temperature (GLDAS) | 4.53 |
| | D04 | geophysical consistency | rise in soil moisture without precipitation (in situ) | 0.28 |
| | D05 | geophysical consistency | rise in soil moisture without precipitation (GLDAS) | 0.26 |
| | D06 | spectrum based | spikes | 0.20 |
| | D07 | spectrum based | negative breaks (drops) | 0.02 |
| | D08 | spectrum based | positive breaks (jumps) | 0.01 |
| | D09 | spectrum based | constant low values following negative break | 0.40 |
| | D10 | spectrum based | saturated plateaus | 2.01 |
| Soil temperature | C01 | range verification | soil temperature $< -60°$C | 0.07 |
| | C02 | range verification | soil temperature $> 60°$C | 0.17 |
| Soil surface temperature | C01 | range verification | soil surface temperature $< -60°$C | 0.01 |
| | C02 | range verification | soil surface temperature $> 60°$C | 0.09 |
| Air temperature | C01 | range verification | air temperature $< -60°$C | 0.03 |
| | C02 | range verification | air temperature $> 60°$C | 0.04 |
| Precipitation | C01 | range verification | precipitation $< 0$ mm$h^{-1}$ | 0.08 |
| | C02 | range verification | precipitation $> 100$ mm$h^{-1}$ | 0.24 |
| Soil suction | C01 | range verification | soil suction $< 0$ kPa | 0.26 |
| | C02 | range verification | soil suction $> 2500$ kPa | 0.00 |
| Snow water equivalent | C01 | range verification | snow water equivalent $< 0$ mm | 20.37 |
| | C02 | range verification | snow water equivalent $> 10000$ mm | 0.01 |
| Snow depth | C01 | range verification | snow depth $< 0$ mm | 19.23 |
| | C02 | range verification | snow depth $> 10000$ mm | 0.00 |

isolated events that do not occur over an extended period of time and thus appear less frequent in the flagging statistics. The

relatively large number of spikes for the network SNOTEL of 0.4% is due to some extremely noisy time series.



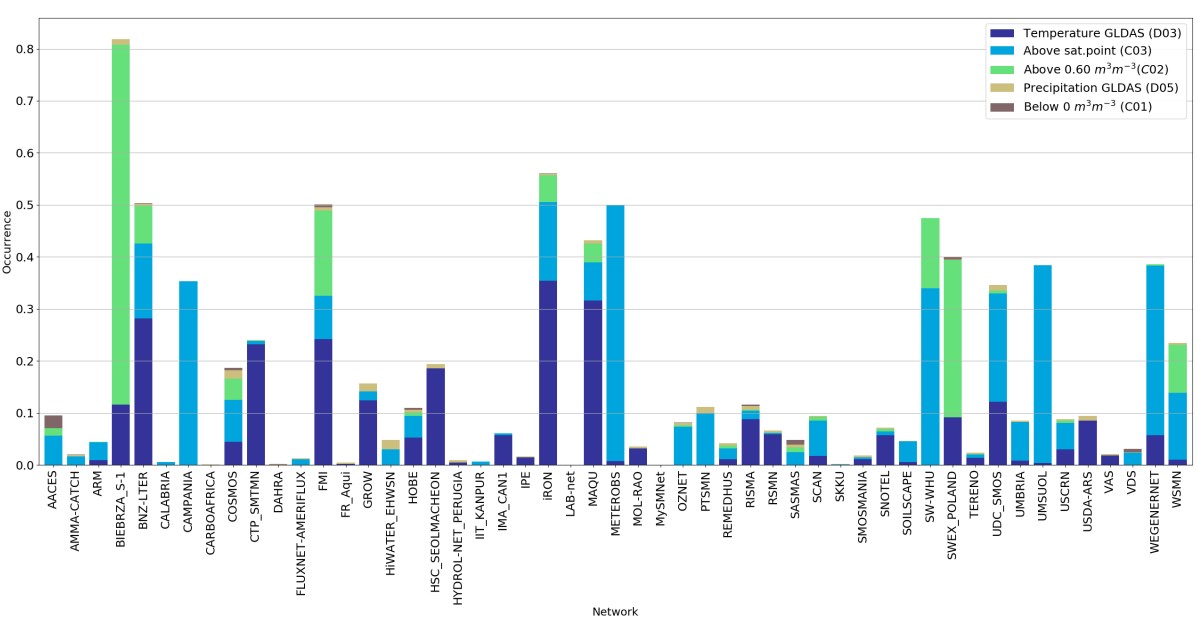

**Figure 4.** Fractions of geophysical dynamic range and consistency quality flags per network. Note that, in principle, the cumulative sum of fractions can be >1 as occurrence of each flag ranges between 0 and 1.

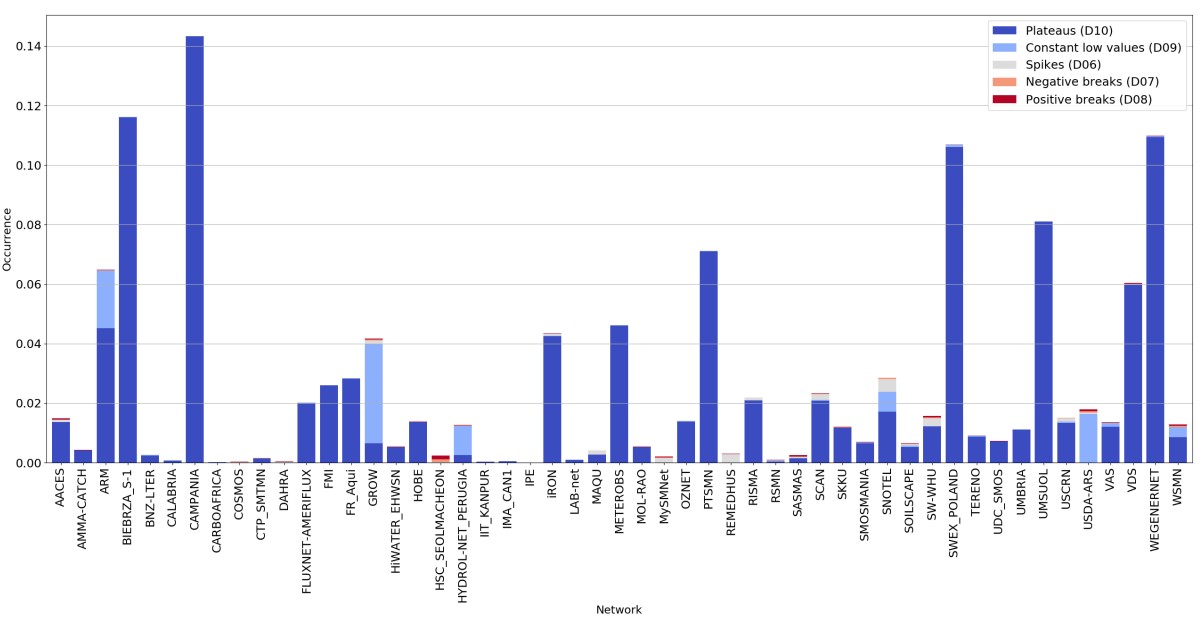

**Figure 5.** Fractions of spectrum-based quality flags per network. Notice that, in principle, the cumulative sum of fractions can be >1 as occurrence of each flag ranges between 0 and 1.





**Table 4.** Global validation results of surface soil moisture of ERA and ESA CCI Soil moisture against ISMN (masked and unmasked for quality flags) for the period 2001-2019; the results for ESA CCI were produced with the QA4SM validation service (https://qa4sm.eu/ Results accessible at: doi.org/10.5281/zenodo.4288919 (absolute values - no flags); doi.org/10.5281/zenodo.4288921 (absolute values - with flags); doi.org/10.5281/zenodo.4288915 (anomalies - no flags); doi.org/10.5281/zenodo.4288913 (anomalies - with flags). ).

| Type | Mask | # time series [$\bar{x}$] | R Pearson [$\bar{x}$] | R Spearman [$\bar{x}$] | ubRMSD [$\bar{x}$] |
|---|---|---|---|---|---|
| ERA5 absolute values | None | 7822 | 0.53 | 0.55 | 2.06 |
| | Flagged values | 5878 | 0.59 | 0.60 | 0.08 |
| ERA5 anomalies | None | 7822 | 0.44 | 0.46 | 1.21 |
| | Flagged values | 5878 | 0.48 | 0.49 | 0.05 |
| CCI absolute values | None | 1178 | 0.46 | 0.47 | 2.98 |
| | Flagged values | 1115 | 0.46 | 0.46 | 0.09 |
| CCI anomalies | None | 1178 | 0.34 | 0.34 | 1.60 |
| | Flagged values | 1115 | 0.36 | 0.35 | 0.06 |

## 3.3 Effect of flagging on applications

For a selected ISMN site (SCAN, Mayday station), Dorigo et al. (2013) showed that masking flagged values has a small but positive impact on the validation of GLDAS-Noah v1 modelled surface soil moisture and the remotely sensed VUA-NASA AMSR-E soil moisture product. Here, we performed a more extensive assessment of the impact of excluding automatically
detected spurious observations by the revised flagging methods (Section 3.1) by using ISMN observations available in the period 2001-2019 to validate both ERA5 top layer (0-0.07 m) water content (Hersbach et al., 2020) and ESA CCI Soil Moisture (v5.2; Gruber et al. (2019b), Dorigo et al. (2017), Gruber et al. (2017)). While the impact of flagging is positive for temporal agreement (R Pearson and R Spearman) between the ISMN and ERA5, the effect is negligible for ESA CCI (Table 4). On the other hand, the ISMN flagging reduces the ubRSMD for both comparisons. The benefit of excluding spurious values is also
obvious in Figure G1a where points are located below the 1:1 line. Again, the benefit is less clear for ESA CCI (Figure G1c). Although validations of a satellite and a model-based are not directly comparable, one reason for the different impact is that the ESA CCI retrievals were already flagged in the production process for values outside valid geophysical range, inconsistencies, dense vegetation, freezing and snow-cover, while this is not the case for the ERA5 model data. Consequently, a positive effect of the ISMN flags is more effective for data sets that were not *a priori* masked than for data sets that were already filtered for
spurious observations.





### 3.4 Other quality indicators

The automated quality control algorithms offer insight into the quality of the respective measurement time series but not necessarily of the usability of the data sets for specific applications. Gruber et al. (2013) adopted a triple collocation approach using soil moisture data from the ISMN, ESA CCI SM v0.1 (Liu et al., 2012b), and ERA-interim land surface model (Dee et al., 2011) to characterise the representativeness errors of ISMN data for coarse-scale (∼25 km) use. Here, we apply a similar approach to estimate the representativeness errors of the ISMN data of all networks with sufficient sampling in the period 2001-2019 for application at the coarse scale. ESA CCI SM passive soil moisture (v5.2, Gruber et al. (2019b), Dorigo et al. (2017)) and top-layer ERA5 volumetric soil water content (Hersbach et al., 2020) are used to complement the triplets. Spatial collocation is carried out using a nearest neighbor method with a maximum distance of 30 km, while the temporal collocation uses a maximum time difference of 1:20 h between the triplets. An exception was solely made for the PBO-H2O network because its observations are provided daily at 12:00 UTC, while ESA CCI SM is given at 00:00 UTC daily. All measurements that cover at least partly the 0.00-0.07 m depth interval are used.

The results for different networks are quite diverse (Figure 6). For example, the spread of errors is relatively large for OR-ACLE but small for others (e.g., AMMA-CATCH or SKKU). The median errors per network vary between 0.03 and 0.05 $m^3 m^{-3}$ with some outliers in both directions. Note that the triple collocation analysis estimates the combined representativeness and sensor errors, although the latter are assumed small comapred to the natural spatial-temporal variability Gruber et al. (2013).

There is a clear trend of decreasing mean errors with increasing sensor depths (Figure 7a), which is likely due to a reduction of high-frequency signals and sensor perturbations with depth. Note, that this decreasing trend is observed despite the increasing discrepancy in depth support between the ISMN data and the two surface soil moisture data sets (ERA5 and ESA CCI SM). Thus, theoretically, in situ representativeness errors are expected to be even lower than computed. A potential explanation for the slight increase in the median error for the deepest layer (100-255 cm) may be small sample size and the poor soil parameterisation of the land surface model at this depth. Similar patterns were observed by Gruber et al. (2013).

Concerning the sensors used, there is a large spread in computed representativeness errors for TDR and capacitance sensors. While the costly hygrometric sensors have the lowest mean error, the mean error of resistance probes is the highest (Figure 7b). Since cosmic ray and GNSS/GPS reflectometry sensors integrate over larger horizontal and vertical domains, one would expect lower representativeness errors for these sensors compared to the point observations. However, this is not confirmed by our triple collocation results. Possibly, the advantage of the larger spatial support of these systems is counteracted by their lower signal-to-noise ratio. Error information at the site-, sensor-, or data set level is currently not routinely available for the stations in the ISMN, but would be required for a proper weighting of individual stations in large-scale validations (Gruber et al., 2018, 2019b).



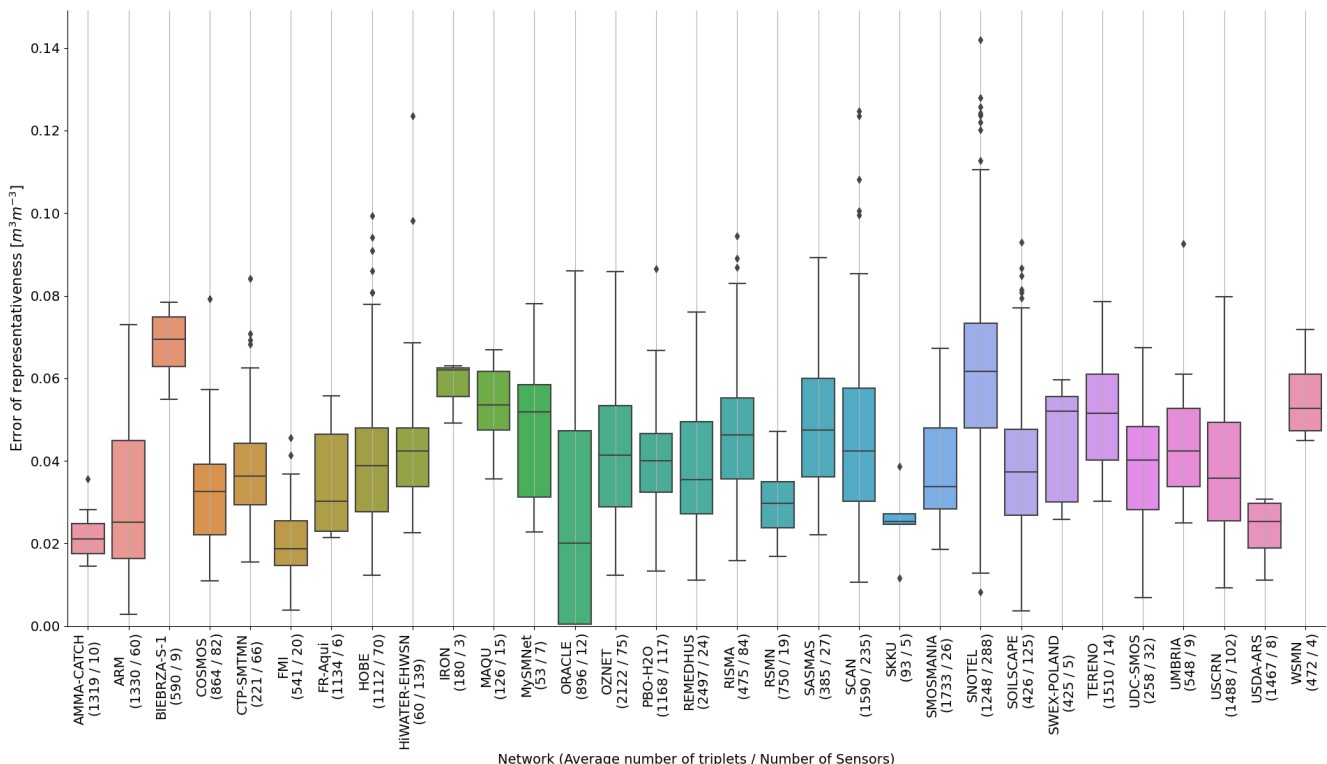

**Figure 6.** Representativeness errors of ISMN networks calculated with triple collocation analysis using top-layer ERA5 volumetric soil water and ESA CCI SM v05.2 passive product. Values in brackets show the average number of triplets per time series and the total number of sensors, respectively, for each network. Following Dorigo et al. (2010) we only used triplets with a Spearman correlation >0.2 between each respective data set pair in the calculation.

## 4 Impact of the ISMN on Earth system sciences

### 4.1 User uptake

As mentioned earlier, over 3200 users have registered to the ISMN while 20-30 new users register each month. Most users
are based in the US, China, India and Europe (Figure 8). When asked for the intended use of the data, the four main GEO benefit areas are water, disaster, agriculture, and climate sectors, all with a similar share between 15-20%. Most users come from non-profit organisations (32%), research organisations (30%), and higher or secondary education (28%). Only few users come from public bodies or private companies (both 5%).





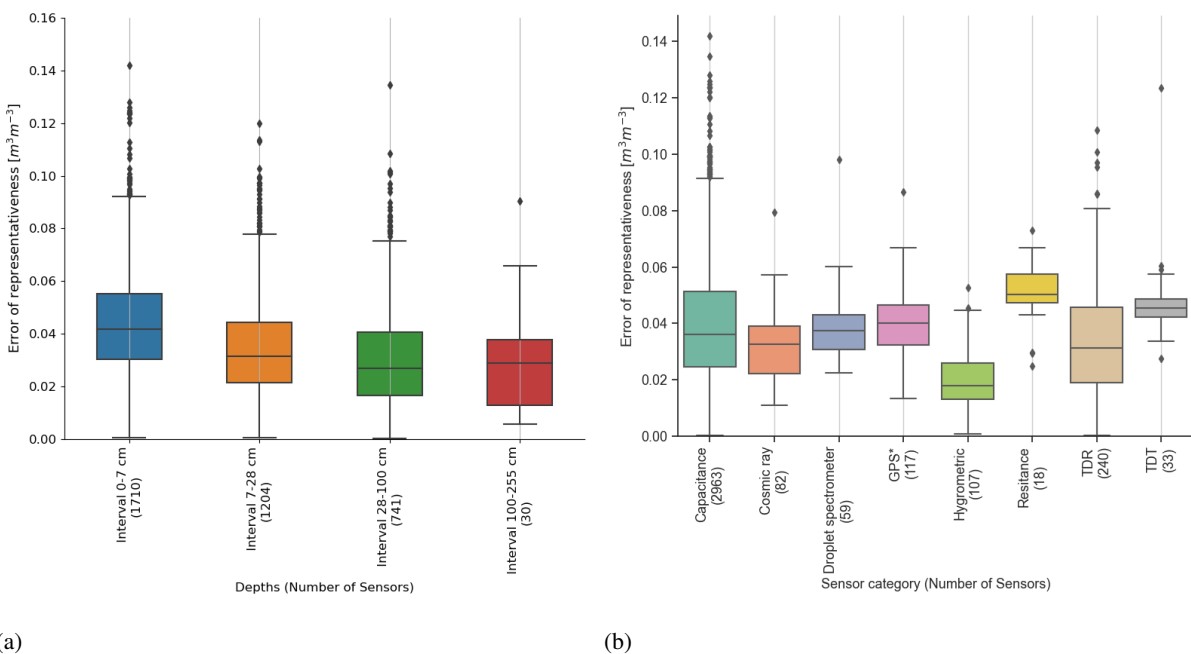

(a)            (b)

**Figure 7.** Representativeness errors for different sensor depths (a) and sensor types (b) derived with triple collocation. Summary of triple collocation characterization of random errors per sensor type. Values in brackets show the total number of sensors for each depth interval and type respectively.

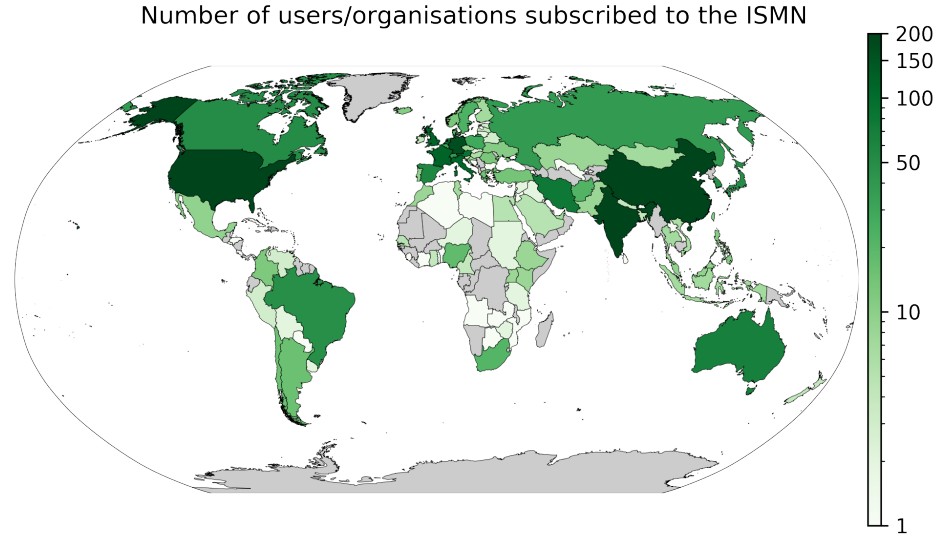

**Figure 8.** Number of users per country (status December 2020).





**Table 5.** Overview of purposes ISMN data are used for in scientific studies (status September 2020; n=288).

| Purpose | % |
| --- | --- |
| Satellite Validation | 53.1 |
| Model development and validation | 14.8 |
| Meteorological applications | 8.9 |
| Drought monitoring | 4.2 |
| Other applications | 19.0 |

The large uptake of the ISMN for soil moisture studies is particularly due to the simplicity of accessing and using multiple
data sets from a wide variety of networks. Initially, the ISMN was established to facilitate the calibration and validation of
SMOS-based soil moisture products (Dorigo et al., 2011b) and still, satellite soil moisture calibration, validation, and algorithm
improvement are the primary applications served by the ISMN (Table 5). In the following, we discuss the major purposes the
ISMN has been used for, based on a comprehensive review of all studies that used and cited the ISMN in peer-reviewed
scientific publications.

## 4.2 Satellite product evaluation and development

### 4.2.1 Scientific studies

Soil moisture measurements from the ISMN have been widely used as reference data set for the development and evaluation of
satellite soil moisture products, mostly global coarse-scale surface soil moisture products (Table F1; n=190). Although initially
the goal was to support algorithm development and validation of the SMOS satellite, which was indeed the case for 79 studies,
many other satellite missions profited from the ISMN data, most notably ASCAT (n=45),AMSR-E (n=44), SMAP (n=39),
and ESA CCI (n=22). Fortuitously, the data have also been discovered for the evaluation of soil moisture products from less
used sensors, including the Chinese Feng-Yun 3B, HY-2 and Gaofen-1 satellites (Parinussa et al., 2014a, 2018; Zhao et al.,
2014; Xing et al., 2017), MSG SIVIRI (Leng et al., 2015, 2017), MODIS (Gumbricht et al., 2017; Gumbricht, 2018), Aquarius
(González-Zamora et al., 2016), and Landsat (Zhao et al., 2017; Pradhan, 2019).
Recently, the ISMN was recognised as validation source for testing algorithms to derive soil moisture from Global Naviga-
tion Satellite Systems (GNSS; e.g., Kim and Lakshmi (2018); Chew and Small (2020). Similarly, there is an increasing trend
in the use of ISMN data for the validation of novel high-resolution satellite soil moisture products, which either downscalfe
coarse-resolution products through the use of other finer resolution satellite or ancillary data (e.g., Sheng et al. (2019); Hel-
gert and Khodayar (2020)), or directly derive soil moisture from high-resolution Synthetic Aperture Aperture satellites like
Sentinel-1 (e.g., Rodionova (2019b); Foucras et al. (2020)). It should be noted that the ISMN and its contributing networks
are mostly designed for analysing time series, thus lacking reference data to assess spatial patterns in the data, particularly in
high-resolution products (de Jeu and Dorigo, 2016).





Soil moisture measurements from the ISMN have been used as training set for various data-driven approaches. In situ observations were ingested into machine learning frameworks together with several ancillary predictor variables, either to
simulate soil moisture at very high spatial resolution (Zappa et al., 2019) or to create long-term records (O and Orth, 2020). Large-scale monitoring networks are necessary to build reliable models for spatially-wide analysis, while dense networks are ideal for accurate localized models (Senanayake et al., 2019; Abbaszadeh et al., 2019).

Usage-oriented evaluation studies have focussed on the intercomparison of multiple coarse-scale satellite products, using the ISMN data as a reference, either to select the best performing sensor for a specific application or geographic region (e.g.,
Beck et al. (2020); Karthikeyan et al. (2017)) or to combine them in an optimal way to build a superior product (e.g., Liu et al. (2011); Hagan et al. (2020)).

Since direct satellite observations of soil moisture are only possible for the surface layer, most studies concentrate on the evaluation of surface soil moisture. Yet, various studies also focus on products derived from surface soil moisture that represent moisture in deeper (root-zone) layers, either through exponential filtering (e.g., Paulik et al. (2014); Tobin et al. (2017)),
empirical models (Pradhan, 2019), or by assimilating them in land surface models (e.g., Pablos et al. (2018); Blyverket et al. (2019)). Occasionally, the ISMN is used for the evaluation of other satellite data sets, e.g. soil temperature data to validate freeze/thaw state (Rautiainen et al., 2016; Hu et al., 2019).

### 4.2.2 ISMN as part of operational services

Because of its operational nature and advanced quality control procedures, the ISMN has been identified as the primary ref-
erence data source for future operational validation systems for global satellite-based soil moisture products (**?**). But already today, the ISMN is integrative part of several operational satellite soil moisture production chains:

- In 2005, the **Satellite Application Facility on support to operational Hydrology and water management (H SAF)** started to operationally produce and validate precipitation, soil moisture, and snow products from satellites operated by EUMETSAT (Rinollo et al., 2013). ISMN data are used for calibration and validation of various soil moisture products
produced in H SAF from the Metop Advanced Scatterometers, including global climate data records and near-real-time products of surface soil moisture, and a root-zone soil moisture product over Europe.

- The **Climate Change Initiative of the European Space Agency (ESA CCI)** uses data from the ISMN to assess each year the quality of new soil moisture climate data record releases and their improvements with respect to forerunner versions (Gruber et al., 2019b; Dorigo et al., 2015). Within ESA CCI, ISMN data were also used to quantify the spatial
representativeness of ESA CCI satellite and climate model data sets (Nicolai-Shaw et al., 2015b).

- **C3S** produces authoritative, quality-assured climate data records of soil moisture and other essential climate variables (ECVs). The satellite soil moisture products produced within C3S are routinely updated every 10 days with the latest available satellite observations. C3S uses the ISMN soil moisture data in combination with the metadata provided to categorise product performance per land cover and climate type (Scanlon et al., 2019).





– **Copernicus Global Land Service (CGLS)** produces within 1-2 days after satellite overpass soil moisture data sets from
Sentinel-1 and from a combination of Sentinel-1 and ASCAT (SCATSAR) (Bauer-Marschallinger et al., 2018, 2019).
The latter propagates the surface observations to deeper layers by means of the so-called soil water index (SWI). Both
products are provided at 1 km resolution for Europe. ISMN data are used for validating moisture in surface and deeper
soil layers.

Recently, the **Quality Assurance for Soil Moisture (QA4SM)** service (https://qa4sm.eu) was initiated to bring together
validation methodologies, community protocols, reference data, and satellite observations to evaluate and intercompare soil
moisture data products in a coherent, standardised, and transparent way. In situ data sourced from the ISMN are integral part
of this validation system (Scanlon et al., 2019; Gruber et al., 2020).

### 4.3    Model development and validation

In situ measurements are the most important reference source when assessing the performance of land surface models, reanal-
yses products, and hydrological models. Although also satellite observations are a valuable validation source (e.g., Szczypta
et al. (2014)), these measure only the upper 5 cm of the soil, and hence do not allow for the validation of deeper layers. Besides,
state-of-the-art reanalysis products like ERA5 (Hersbach et al., 2020)) assimilate satellite soil moisture observations, so that
for these products in situ data remain the only truly independent validation reference. Since the sampling rate (hourly) of the
ISMN observations is generally higher, and their spatial support lower than that of most models (Reichle et al., 2004), model
evaluation is in principle not limited by the spatial and temporal resolution of the in situ data, although representativeness
issues often remain.

    In particular for global assessments, the availability of harmonised data over multiple networks makes the ISMN a preferred
reference source over data from individual networks (Xia et al., 2019). Table F2 shows that many well-established state-of-
the-art land surface model and reanalysis products have been evaluated against data from the ISMN, not only by users of these
products (e.g. (Li et al., 2020b), but also by the developer teams themselves, e.g., Reichle et al. (2017) for MERRA2, Albergel
et al. (2020) for LDAS-Monde, and Balsamo et al. (2015); Fairbairn et al. (2019) for various ECMWF products.

    Also a suite of new products have been assessed against the ISMN, including multi-model ensembles (Schellekens et al.,
2017; Cammalleri et al., 2015), data-driven evaporation models (Martens et al., 2017; Lievens et al., 2017) and statistical
infiltration models Pal and Maity (2019). As for remote sensing products, a trend towards higher-resolution regional to global
model-based products can be observed. Among others, ISMN data have been used to validate new high-resolution reanalysis
products over the United States (McDonough et al., 2018) and Europe (Naz et al., 2020). Apart from soil moisture also soil
temperature data have been drawn from the ISMN for model validation purposes (e.g., Wang et al. (2016); Albergel et al.
(2015)).

Besides the evaluation of hydrological or land surface model improvements, the ISMN has also frequently served model
development in a more fundamental way. For example, Hartmann et al. (2015) developed a large-scale karstic groundwater
recharge model over Europe and the Mediterranean and calibrated and evaluated this model with observations of actual evap-





otranspiration from FLUXNET (Baldocchi et al., 2001) and soil water content data from the ISMN. Calvet et al. (2016) used soil moisture and temperature data from the SMOSMANIA network contained in the ISMN to derive pedotransfer functions

for soil quartz fraction in southern France. Pal et al. (2016) developed a statistical model to estimate vertical soil moisture profile using SCAN data of the ISMN as source, while on a similar note, Shin et al. (2018) developed a non-parametric evolutionary algorithm to predict soil moisture dynamics using ISMN data over Oklahoma and Illinois. Similarly, Jalilvand et al. (2018) estimated the drainage rate from surface soil moisture drydowns from ISMN data. A bridge between soil moisture and vegetation dynamics was made by Sawada (2018), who used ISMN data to validate a new ecohydrological land reanalysis to

better simulate the link between sub-surface soil moisture and vegetation dynamics. Finally, Brocca et al. (2015) developed a water-balance approach to estimate rainfall from soil moisture observations based on reverse modelling and evaluated this at several ISMN sites in Europe. Later refinements of this approach were also evaluated against data from the ISMN (Hoang and Lu, 2019).

## 4.4 Meteorological applications

Soil moisture from the ISMN has often been used to validate the land surface representations of meteorological forecasting models. However, as meteorological forecasts often rely on the latest generation of land surface models, in practice there is often no strict distinction between meteorological and land surface model development as described in the previous section. Notable examples are the various generations of TESSEL models used both in the Integrated Forecasting Systems and reanalysis products of ECMWF, the development of which greatly profited from soil moisture and temperature data from the ISMN

(Albergel et al., 2012a, 2015). Dirmeyer et al. (2016) assessed the skill and soil moisture memory effects of various weather and climate models with ISMN data over the United States. Similarly, Angevine et al. (2014) assessed the soil moisture skill of the Weather Research and Forcasting Model (WRF) (Table F2).

Several studies have used the ISMN data to assess the forecast skill or new implementations of numerical weather prediction models. For example, de Rosnay et al. (2019) and Rodriguez-Fernandez et al. (2019) used in situ soil moisture observations

from several ISMN networks to validate the impact of assimilating SMOS brightness temperatures and soil moisture, respectively, to predict soil moisture up to 5 days ahead. Similarly, Lin and Pu (2019, 2020) used ISMN data to examine the impact of MAP soil moisture assimilation in WRF for near-surface short-range weather forecasts. Boussetta et al. (2015) used over 500 ISMN sites to assess the impact of assimilating surface albedo and vegetation states from satellite observations on numerical weather prediction.

From a more methodological, land-atmosphere perspective, Lee (2018) used the ISMN data to study the role of soil moisture in triggering rainfall over West Africa. Vice versa, Zhang et al. (2019c) studied the role of rainfall on soil cooling using data from the SMOSMANIA network in southern France.

## 4.5 Drought monitoring

In a drought monitoring context, ISMN data have frequently been used in a "convergence of evidence" approach in combination

with other drought-related variables or indicators. For example, Scaini et al. (2015) compared the variability of in situ soil





moisture measurements from the ISMN, SMOS surface soil moisture, and two drought indices based on climatic information to study droughts in Spain. Mu et al. (2019) used VNIR satellite data and soil moisture distributed by the ISMN to monitor drought in the southern United States. On a more technical level, Gruber et al. (2018), assessed the use of spatially sparse ISMN in combination with a continuous model for operational agricultural drought monitoring over the United States.

ISMN data have also been used to classify new and more "traditional" drought indices, such as the Standardized Precipitation (Evaporation) Index and the Palmer Drought Severity Index (Vicente-Serrano et al., 2012; Krueger et al., 2019). Sadri et al. (2020) used the ISMN data from the RISMA network to assess a global near-real-time soil moisture index monitor for food security using SMOS and SMAP data. Chen et al. (2019) used precipitation distributed through the ISMN to evaluate a drought index derived from Sentinel-2 in Spain. Due to relatively good coverage over Europe, Cammalleri et al.

(2015) used ISMN data to assess various model soil moisture products as input to the European Drought Observatory (EDO, http://edo.jrc.ec.europa.eu). Also Mishra et al. (2018) used data from the ISMN to assess a suite of land surface models used to reconstruct drought events in India since the mid 1900s.

## 4.6   Other applications

ISMN data have been used for various other purposes, going beyond what the ISMN was originally developed for. In addition
to supporting satellite and land surface model soil moisture product development, the ISMN has played a fundamental role in the validation of a wide range of hybrid observation-based products, including a Soil Moisture Saturation Index (Campo et al., 2011), apparent thermal inertia surface estimates (Notarnicola et al., 2012), data-driven surface and root-zone soil moisture predictions (Kornelsen and Coulibaly, 2014; Manfreda et al., 2014; Wang et al., 2020a), improved satellite albedo products (Liu et al., 2014), and estimates of effective permittivity and brightness temperature of organic soils Park et al. (2019).

The ISMN has frequently been used to asses the impact of assimilating satellite observations into hydrological models (Khaki et al., 2019; Nair et al., 2020a; Gruber et al., 2015; Shin et al., 2016; Li et al., 2020b; Wang et al., 2020b), land surface models (Nair and Indu, 2016; Zhao and Yang, 2018; Nair et al., 2020b) and carbon models (Scholze et al., 2016). On a more methodological level, Zhang et al. (2019a) assessed a new data assimilation scheme against ISMN observations. Gruber et al. (2018) even assimilated spatially sparse ISMN observations directly into a spatially continuous land surface model over the
Continental US, and tested the preconditions (e.g., requirements on signal-to-noise ratio and number of sites) for having a significant positive impact.

The wealth of data covering a wide range of surface and climate conditions has been frequently exploited to study soil water dynamics at various spatial and temporal scales and their (climatic) drivers (Brocca et al., 2014b; Hirschi et al., 2014; Ojha et al., 2014; Qin et al., 2018; Kumar et al., 2019a; Tian et al., 2019a; Nicolai-Shaw et al., 2015b; Dong et al., 2020; Wang et al.,
2017; Verrier, 2020; Deng et al., 2020a), flood dynamics (Esposito et al., 2018), or the impact of soil moisture on plant growth (Hottenstein et al., 2015).

The ISMN has been used to develop and test new statistical validation approaches (Zwieback et al. (2013), Gruber et al. (2016), Afshar et al. (2019)). Finally, the standards and quality control procedures adopted by the ISMN have served as a guideline for establishing and benchmarking new networks ((Kim et al., 2019; Skierucha et al., 2012b, a; Petropoulos and





McCalmont, 2017)), soil moisture metadatabases (Liao et al., 2019; Xia et al., 2015)), or validation services (Kumar et al., 2012), and to assess or improve alternative sensing techniques and constellations (Xaver et al., 2020; Kapilaratne and Lu, 2017; Nguyen et al., 2017; Mahecha et al., 2017).

## 5 Challenges and opportunities

### 5.1 Scientific challenges

#### 5.1.1 Diversity of measurements

The in situ measurements of soil moisture data in the ISMN have been collected by a large variety of sampling techniques. The early networks contained in the ISMN (e.g., RUSWET, CHINA and MONGOLIA) are based on gravimetric sampling, which is still considered the most accurate approach (Romano, 2014). However, it is labor-intensive and invasive and thus measurements are infrequent (weekly or even coarser sampling), every time being taken from a slightly different location, may
contain systematic errors between sampling dates.

Nowadays, the most commonly used techniques for systematic in situ sampling are based on the contrasting dielectric properties of soil and water and comprise time domain reflectometry (TDR) and frequency domain reflectometry (FDR) (Robinson et al., 2008), e.g., AMMA-CATCH, BNZ-LTER, CARBOAFRICA, FMI, GTK, HYDROL-NET-PERUGIA, LAB-net, OR-ACLE, OZNET, SASMAS, SWEX-POLAND, UDC-SMOS, WSMN, UMSOL networks). In particular, capacitance sensors
based on FDR are becoming more and more widespread because of their lower cost compared to TDR sensors, despite their lower accuracy (Romano, 2014; Brocca et al., 2017). However, great technical improvements are being achieved with capacitance sensors, thus improving their reliability. Yet, TDR and FDR techniques only provide point measurements, i.e. they are representative of small volumes of soil.

Slightly larger soil volumes (diameter of 15 to 60 cm) are observed with the neutron scattering method, where the density of
thermal neutrons produced by scattering of fast neutrons on soil hydrogen can be related can be related to soil moisture through a calibration curve (Gardner and Kirkham, 1952; Romano, 2014) (e.g., IOWA network). The cosmic-ray method has been developed based on similar principles of neutron scattering, (Zreda et al., 2012, 2008). Cosmic-ray neutron sensors (CRNS) are located on or above the soil surface and measure and count the number of cosmogenic neutrons in air above the land surface and that are in equilibrium with the soil. CRNS measurements are representative of significantly larger volumes compared to
neutron probes (i.e., radius of a few 100 metres and depths between 0.12-0.70 m, depending on the soil wetness), and thus can be used to bridge the scales between point observations and coarse resolutions of satellite observations and models. CRNS measurements, which are used in the COSMOS network, are sensitive to vegetation and have relatively high noise levels. To reduce this noise, the data are resampled to daily mean values.

Also the measurements of the PBO_H2O network, which uses Global Positioning System (GPS) receivers, can be used to
bridge the scales between point and satellite sensors. GPS sensors, initially used for geophysical and geodetic applications, have been found well suited to measure soil moisture (Larson et al., 2008). The GPS signals are representative for an area





of approximately 300 m$^2$ and are L-band, hence making them ideal for comparison against satellite missions like SMAP and SMOS (Larson et al., 2008).

As shown, all measurement techniques have their strengths and limitations Bogena et al. (2015); Dorigo et al. (2011b),
which complicates their combined, bulk use. Depending on the application and process scales, the user needs to carefully consider which networks to use or exclude and how to interpret the results obtained. A task of the ISMN could be to translate community-based guidelines (Gruber et al., 2020; Montzka et al., 2020) to recommendations for the use of individual datasets.

### 5.1.2 Spatial representativeness and scaling

Soil moisture is highly variable in space as a result of complex interactions between soil characteristics, topography, vegetation,
and meteorological conditions. Depending on the spatial scale considered, the dominant controlling factor(s) can be different (Crow et al., 2012; Western et al., 2002). Hence, validation of satellite-derived products is hindered by the spatial mismatch between ground observations and satellite footprints (Gruber et al., 2020; Molero et al., 2018; Gruber et al., 2013). Ideally, to reduce the representativeness error of in situ references, enough stations should be deployed within a satellite footprint to develop robust areal soil moisture estimates (Brocca et al., 2007; Famiglietti et al., 2008; Colliander et al., 2017). However,
this is a costly solution and therefore only a limited number of sites provide such setup (Crow et al., 2012; Colliander et al., 2017). In the ISMN the networks VDS, BIEBRZA_S-1, RSMN, UDC_SMOS, LAB-net, FMI, USDA-ARS have been set up in this way. When available, in situ stations within the same satellite pixel should be averaged, either through arithmetic mean or weighted average. Higher weights should be given to stations expected to be more representative of the satellite grid average, e.g., by using Voronoi diagrams (Colliander et al., 2017), the inverse footprint method (Nicolai-Shaw et al., 2015a),
the time stability concept (Vachaud et al., 1985), or landscape properties such as land cover and/or soil texture (Bircher et al., 2012). Alternatively, the triple collocation (TC) can be used to quantify and correct for spatial sampling errors of in situ stations (Miralles et al., 2010; Gruber et al., 2013). The resulting pixel-scale soil moisture ground reference, i.e., the averaged value from dense networks with several stations per satellite grid or the original time series in the case of sparse networks with a single station per pixel, should undergo a statistical rescaling (Gruber et al., 2020). Indeed, a direct comparison of in situ and
satellite products would be subject to representativeness errors, which may dominate the total soil moisture retrieval errors (Chen et al., 2017; Gruber et al., 2013; Molero et al., 2018). Rescaling accounts for systematic representativeness errors arising from different spatial resolution and different vertical measurement support, i.e., penetration depths of microwave sensors and in situ sensor placement depths (Gruber et al., 2013, 2020) but does not correct for random representativeness errors. One way to address this is the triple collolacation described above. Also, systematic representation errors may have a time-varying (e.g.,
seasonal) component, which, unless explicitly accounted for, may lead to temporally aliased results.

### 5.1.3 Integration of low-cost sensors

The development and use of low-cost sensing technologies, especially in the environmental sciences, has seen a pronounced increase during the last decade. Such a rise is driven by several factors, e.g., the reduced cost of micro-controllers, electric components, and sensors (Mao et al., 2019). Even though a rich variety of low-cost soil moisture sensors based on different





measurement principles has been developed (Chawla et al., 2019; González-Teruel et al., 2019; Kumar et al., 2016), capacitance sensors gained most popularity because they are relatively inexpensive, easy to operate, and provide reliable observations (Kojima et al., 2016).

The considerably lower cost of these sensors compared to traditional probes makes them suitable for high-density and/or large-scale monitoring of soil moisture. The possibility to map soil moisture (as well as other environmental variables) with an unprecedented spatial coverage can generate new insights into its dynamics and create new opportunities. For instance, 510 high-density networks of low-cost sensors can be used to reduce spatial representativeness errors by providing numerous observations within a satellite footprint (Teuling et al., 2006). Similarly, one can deploy a temporary low-cost sensors network to identify the most suitable location(s) for long-term monitoring. Such locations could then be equipped with professional sensors, while moving the low-cost network to other sites (Zappa et al., 2019). Another exciting opportunity offered by low-515 cost sensors is the deployment of networks in low-income countries.

However, a number of practical challenges arise when integrating low-cost sensors in the ISMN. A key aspect to consider is the lifetime: depending on the robustness of the sensor (both of electronic components, such as micro controllers, and sensor housing), reliable measurements can be recorded for a period ranging from a few months to years (Xaver et al., 2020) but even be shorter in extreme environments. Another aspect, particularly affecting automation, is data storage and transmission. Some 520 low-cost sensors allow for wireless communication with a main server ((Bogena et al., 2007), (Majone et al., 2013)), while other sensors have a limited internal storage and data should be collected persistently over time (e.g., every 80 days, Xaver et al. (2020)). Furthermore, it is necessary to assess their accuracy and robustness (Castell et al., 2017). Therefore, low-cost sensors should always undergo thorough evaluation to i) quantify the agreement of low-cost sensor measurements with gravimetric samples and/or professional probes, ideally considering a wide range of soil and climatic conditions, ii) assess the inter-sensor 525 variability, and iii) test the suitability for usage in field conditions (Domínguez-Niño et al., 2019; Kizito et al., 2008; Mittelbach et al., 2012; Adla et al., 2020).

### 5.1.4 Integration of citizen observations

Citizen science is defined as the involvement of non-experts in collecting data (Bonney et al., 2009). Crowd-sourced measurements of soil moisture are now possible because of the development of low-cost sensors (Section 5.1.3). Crowd-sourcing has 530 the potential to overcome some of the most challenging issues of soil moisture monitoring, such as the use of many sensors to address scaling issues, and the fact that the observations can be carried out anywhere, as long as there are citizens willing to collect the data.

An outstanding example of a citizen science project focusing on soil moisture is the GROW Observatory (growobservatory.org). Within GROW, thousands of low-cost sensors have been distributed to farmers, gardeners, and growers across Eu535 rope (Kovács et al., 2019). Focus areas have been identified based on a number of scientific criteria and the presence of active and engaged communities. Within each focus area, covering spatial scales from 20 to 200 km, hundreds of sensors have been distributed. Some sensors served as back-up for potential failures, so that malfunctioning sensors could be promptly replaced enabling long-term continuity of observations. In order to ensure high standards of the measurements, citizens have been





trained in the selection of the correct locations where to install sensors, how to properly install and maintain them through
field manuals, online courses, meet-ups and remote support (Kovács et al., 2019). Overall, more than 6000 sensors were de-
ployed and provided soil moisture measurements across 13 European countries, demonstrating that citizen observatories can
be integrated in EO activities and contribute to validation of remotely sensed products (Zappa et al., 2020).

The integration of citizen observations in the ISMN is challenging for multiple reasons. Crucial is the long-term engagement
of citizens, which needs to be thoroughly addressed from the early stages of designing a citizen observatory. It is necessary to
create long-lasting communities that go beyond the duration of the contributory projects (Grainger, 2017). Successful citizen
observatories have been those where citizens were able to benefit directly from the data collected by others (Fritz et al., 2017).
Additionally, education and resources are essential to boost individual motivation to continuous participation and to sustain re-
newal of membership as natural participation cycles change (Zappa et al., 2020). Even if proper training is provided to citizens,
uncertainty in the quality of the data is the main limitation for the use of crowd-sourced observations in science (Lukyanenko
et al., 2020). The reliability of measurements from individual sensors is unknown, because of, e.g., the selection of unsuitable
locations, incorrect sensor installation, and existence of defective sensors. However, increasing the number of sensors within the
same satellite pixel reduces this uncertainty. Additionally, visual inspection of time series, as well as application of automated
quality flagging controls, such as those developed within the ISMN, could be used to mask suspicious observations (Xaver
et al., 2020). Overall, it has been shown that well-organized citizen science projects can provide trustworthy contributions to
the scientific community (Kosmala et al., 2016; Palmer et al., 2017).

## 5.2 Operational challenges

### 5.2.1 Automation

Part of the contributions to the ISMN (6 networks with over 900 stations) are inserted into the ISMN in NRT. This process
is fully automated, including data downloaded from data providers, harmonisation, flagging, insertion into the database, and
updating metadata tables. However, because of different data recording and handling mechanisms at the provider side and
differences in data sharing mechanisms and policies the ISMN is also partially manually operated and will probably also have
to continue in this way in the future. The differences between the fully automated and the manual approach is confined to data
download and data ingestion into the processing chain. Data harmonisation and quality control is automated for all data sets
(see Figure C1).

A major challenge in the automation process is the enormous heterogeneity of input data formats. Moreover, these change
over time for individual networks, stations, and even sensors, as sensors may fail or the method for data logging is changed.
Thus, error detection and handling is of utmost importance and frequent adaptation of the system required to cover changes in
input data.

A potential way to promote the automated insertion of new data is to allow only data that comply with a strict, yet to be
defined, data standard. At the same time, this may be bear the risk of raising the barrier to contribute to the ISMN too high for
several scientists.





### 5.2.2  Including new networks

Since data sharing with the ISMN is entirely built upon a voluntary basis and data usage is open and free, new networks may be reluctant to join. The ISMN is in contact with several network providers who are happy to collaborate but are restrained by data sharing policies, which does not allow data sharing at all or only after a certain time. Furthermore, for governmentally operated networks it is often not allowed to share data as open-access or it is unclear who can make these decisions.

Solving such issues can be supported by collaborative data hosting facilities like the Global Terrestrial Network - Hydrology (GTN-H; gtn-h.info), who have a strong connection to governmental bodies like the International Centre for Water Resources and Global Change under the auspices of UNESCO, the United Nations Environment Programme, the International Science Council, and WMO.

### 5.2.3  Appropriate recognition of data providers

Not all users correctly cite the ISMN and the involved networks, as stated in the terms and conditions for ISMN data use (ISMN). Proper citation of single networks is important for giving data providers the recognition that is required to convince funding agencies to continue supporting the maintenance and operation of these networks. In the end this does not only affect the networks themselves but also the open-access availability of data through the ISMN as a whole. Some network providers have raised their concerns in this matter and, hence, continuous efforts are needed to maintain a strong visibility of network data providers towards users.

### 5.2.4  Transparency and traceability

The ISMN data collection is constantly evolving: New data records are added and existing ones are extended or, if necessary, reprocessed and corrected. Not only the data, but also the underlying code changes. These updates and any retroactive changes made to the data archive are tracked within the ISMN but not yet readily passed on to the users. The updates can lead to differences in analyses on the user side, e.g., when considering obvious changes such as temporal extensions or new stations, and lead to non-reproducible results. Therefore, any update must be traceable and clearly communicated, requiring a system to track and store these changes. Version tracking and digital object identifiers (DOIs) are ways to identify each database access and therefore allow tracing back to past states of the ISMN. Such a mechanism is require to make the ISMN compliant with the FAIR Data Principles (Wilkinson et al., 2016).

## 6  Conclusions

In this study we reviewed the first decade of operations of the ISMN. Besides satisfactorily fulfilling its initial target, i.e., supporting satellite soil moisture product validation and calibration, many additional more or less foreseen uses have emerged. In addition, an increasing number of services and product development chains have routinely included the use of ISMN data in their operational structure. While the development and operations of the ISMN itself have received continuous financial support

from ESA, all network data sets have been freely contributed by dedicated researchers. To guarantee the availability of these resources for climate and environmental monitoring also for the next decade, we plead with governments and international bodies for systematic funding of both the ISMN and its participating data-providing networks.

To maximise geographic coverage and data usage, the policy of the ISMN has always been to integrate datasets without strict requirements on sensors, sampling protocol, or data quality. The emerging strongly heterogeneous data set characteristics call for far-going quantification and traceability of errors, from sensor calibration to data download and from the point measurement to the spatial-temporal support of the application. Thus, besides expanding its coverage to data-poor regions and landscapes, the ISMN shall focus the next 10 years of operation on developing methodologies that allow observations to qualify as Fiducial

Reference Measurements for soil moisture applications.

*Code and data availability.*

    Upon registration, all data and metadata described in this paper can be downloaded for free from https://ismn.earth. A python package to read and plot the data and metadata ((TUW-GEO/ismn: v0.4) can be accessed on doi.org/10.5281/zenodo.3966399.

An example of output of this package is shown in Figure 1. A python package containing the ISMN quality control procedures for in situ soil moisture is available on GitHub (https://github.com/TUW-GEO/flagit).

*Author contributions.* Wouter Dorigo conceived the manuscript and wrote it together with Irene Himmelbauer, Daniel Aberer, Lukas Schremmer, Ivana Petrakovic, Luca Zappa, and Wolfgang Preimesberger. All others contributed data to the ISMN and provided feedback on the manuscript

*Competing interests.* No competing interests

*Acknowledgements.* The authors greatly acknowledge the financial support provided by ESA through various projects: SMOSnet International Soil Moisture Network (contract No. 4000102722/10/NL/FF/fk and operations contract No. 3-13185/NL/FF/fk); IDEAS+ (contract No. TVUK/AG/18/02082); QA4EO (contract No. TPZV/UK/AG/19/02321). Additional funding for methodological advances has been received from the EU FP7 EartH2Observe project (Contract No. 603608), the EU H2020 GROW project (Contract No. 690199), and the

QA4SM project funded by the Austrian Space Applications Programme 14 (Contract No. 866004). We acknowledge the endorsement of various international bodies, including CEOS, WCRP GEWEX, GCOS, GTN-H, and GEO. We greatly thank all staff members from the participating networks for their continued technical support.





## Appendix A: Network overview

A summary of each network is given in Table A1 while more details are given in the subsequent paragraphs.

**Table A1.** Main properties of networks and data contained in the ISMN.

| Network abbreviation | Location | Number of stations | Start date | End date | Status |
|---|---|---|---|---|---|
| AACES | Australia | 49 | 2005-05-09 | 2010-09-24 | inactive |
| AMMA-CATCH | Benin, Niger, Mali | 7 | 2006-01-01 | 2018-12-31 | running |
| ARM | USA | 35 | 1993-06-29 | 2020-12-10 | running |
| AWDN | USA | 50 | 1997-12-31 | 2010-12-30 | running |
| BIEBRZA_S-1 | Poland | 30 | 2015-04-23 | 2018-12-01 | running |
| BNZ-LTER | Alaska | 12 | 1988-06-01 | 2013-01-01 | running |
| CALABRIA | Italy | 5 | 2001-01-01 | 2012-12-31 | running |
| CAMPANIA | Italy | 2 | 2000-07-27 | 2012-12-31 | inactive |
| CARBOAFRICA | Sudan | 1 | 2002-02-08 | 2010-01-20 | running |
| CHINA | China | 40 | 1981-01-08 | 1999-12-28 | inactive |
| COSMOS | Worldwide | 108 | 2008-04-28 | 2020-03-29 | running |
| CTP_SMTMN | China | 57 | 2010-08-01 | 2016-09-19 | running |
| DAHRA | Senegal | 1 | 2002-07-04 | 2016-01-01 | running |
| FLUXNET-AMERIFLUX | USA | 2 | 2000-01-01 | 2020-02-12 | running |
| FMI | Finland | 27 | 2007-01-25 | 2020-07-04 | running |
| FR_Aqui | France | 5 | 2012-01-01 | 2020-01-01 | running |
| GROW | Europe | 150 | 2017-02-08 | 2019-10-08 | closed |
| GTK | Finland | 7 | 2001-05-16 | 2012-05-29 | running |
| HiWATER_EHWSN | China | 174 | 2012-05-04 | 2012-09-20 | inactive |
| HOBE | Denmark | 32 | 2009-09-08 | 2019-03-13 | running |
| HSC_SEOLMACHEON | Korea | 1 | 2008-01-01 | 2009-01-01 | running |
| HYDROL-NET_PERUGIA | Italy | 2 | 2010-01-01 | 2013-12-31 | running |
| ICN | USA | 19 | 1983-01-03 | 2010-11-21 | inactive |
| IIT_KANPUR | India | 1 | 2011-06-16 | 2012-11-22 | running |
| IMA_CAN1 | Italy | 12 | 2011-08-23 | 2015-12-03 | closed |
| IOWA | USA | 6 | 1972-04-04 | 1994-11-15 | inactive |
| IPE | Spain | 2 | 2008-04-03 | 2020-03-25 | running |
| iRON | USA | 9 | 2012-08-21 | 2019-12-31 | running |
| KHOREZM | Uzbekistan | 7 | 2010-04-17 | 2011-09-10 | inactive |





Table A1 continued from previous page

| Network abbreviation | Location | Number of stations | Start date | End date | Status |
|---|---|---|---|---|---|
| KIHS_CMC | South Korea | 18 | 2019-03-20 | 2019-12-10 | running |
| KIHS_SMC | South Korea | 19 | 2019-03-21 | 2019-12-05 | running |
| LAB-net | Chile | 3 | 2014-07-18 | 2017-11-21 | running |
| MAQU | China | 20 | 2008-06-30 | 2010-07-31 | running |
| METEROBS | Italy | 1 | 2011-10-23 | 2012-05-09 | inactive |
| MOL-RAO | Germany | 2 | 2003-01-01 | 2014-01-01 | running |
| MONGOLIA | Mongolia | 43 | 1964-04-08 | 2002-10-18 | inactive |
| MySMNet | Malaysia | 7 | 2014-05-31 | 2015-12-31 | running |
| ORACLE | France | 6 | 1985-10-18 | 2013-09-09 | running |
| OZNET | Australia | 38 | 2001-09-12 | 2018-08-27 | running |
| PBO_H2O | America, Africa, Asia | 158 | 2004-09-27 | 2017-12-16 | inactive |
| PTSMN | New Zealand | 20 | 2016-10-30 | 2018-11-15 | running |
| REMEDHUS | Spain | 24 | 2005-03-15 | 2020-01-01 | running |
| RISMA | Canada | 23 | 2013-04-24 | 2020-03-25 | running |
| RSMN | Romania | 20 | 2014-04-09 | 2020-06-30 | running |
| RUSWET-AGRO | Former Soviet Union | 212 | 1958-04-08 | 2002-06-28 | inactive |
| RUSWET-GRASS | Former Soviet Union | 121 | 1952-06-08 | 1985-12-28 | inactive |
| RUSWET-VALDAI | Former Soviet Union | 3 | 1960-01-15 | 1990-12-15 | inactive |
| SASMAS | Australia | 14 | 2005-12-31 | 2007-12-31 | running |
| SCAN | USA | 239 | 1996-01-01 | 2020-07-12 | running |
| SKKU | South Korea | 5 | 2014-05-08 | 2014-11-19 | running |
| SMOSMANIA | France | 22 | 2007-01-01 | 2019-01-01 | running |
| SNOTEL | USA | 438 | 1980-10-01 | 2020-07-12 | running |
| SOILSCAPE | USA | 169 | 2011-08-03 | 2017-03-29 | running |
| SWEX_POLAND | Poland | 6 | 2000-01-01 | 2013-05-06 | inactive |
| SW-WHU | China | 7 | 2014-01-12 | 2015-06-03 | running |
| TAHMO | Africa (Sahel Zone) | 70 | 2015-01-01 | 2020-10-31 | running |
| TERENO | Germany | 5 | 2009-12-31 | 2020-07-09 | running |
| UDC_SMOS | Germany | 11 | 2007-11-08 | 2011-11-18 | inactive |
| UMBRIA | Italy | 13 | 2002-10-09 | 2017-12-31 | partially running |
| UMSUOL | Italy | 1 | 2009-06-12 | 2010-09-30 | running |





**Table A1 continued from previous page**

| Network abbreviation | Location | Number of stations | Start date | End date | Status |
|---|---|---|---|---|---|
| USCRN | USA | 115 | 2000-11-15 | 2020-07-11 | running |
| USDA-ARS | USA | 4 | 2002-06-01 | 2009-07-31 | inactive |
| VAS | Spain | 3 | 2010-01-01 | 2012-01-01 | running |
| VDS | Myanmar | 4 | 2017-06-01 | 2020-04-06 | running |
| WEGENERNET | Austria | 12 | 2007-01-01 | 2020-07-11 | running |
| WSMN | UK | 8 | 2011-09-02 | 2016-02-29 | running |

## A1    AACES

AACES stands for the Australian Airborne Cal/val Experiments for SMOS. This campaign network covers a $500 \times 100$ km2 study area located in South-East Australia, covering a variety of topography, vegetation- and climate classes (Rüdiger et al., 2007; Peischl et al., 2012). Measurements of soil moisture, soil temperature, and precipitation were taken between May 2009 and September 2010. The AACES calibration and validation campaigns were a temporary project therefore no further data sets will be available.

## A2    AMMA-CATCH

The AMMA-CATCH observatory gathered data from densely instrumented mesoscale sites in West Africa (Benin, Niger, Mali). The network is devoted to long-term regional monitoring of global change impacts on the critical zone. Height stations in Benin and four stations in Niger of the network are included in the ISMN from 2006 to present, including surface soil moisture and root-zone soil moisture until one-meter depth. For more information, see Galle et al. (2018).

## A3    ARM

The Atmospheric Radiation Measurement (ARM) Program has three soil sensor networks across north-central Oklahoma and southern Kansas in the U.S.: The Soil Water and Temperature System (SWATS) through 2016, and presently the Soil Temperature and Moisture Profile (STAMP) and Surface Energy Balance System (SEBS). The SWATS and STAMP have profiles at 5-8 depths up to 175 cm, while the SEBS measure at 2.5 cm. All sites are co-located with a suite of meteorological and radiative measurements available from ARM.gov (Cook, 2016a, b; Cook and Sullivan, 2018).



## A4 AWDN

The AWDN network is located in Nebraska, United States, and consists of 50 stations. The data sets were collected by the High Plains Regional Climate Center", Data availability is from 1998 to 2010 but varies per station.

## A5 BIEBRZA S-1

Two dense soil moisture sites suited for the validation of high-resolution Sentinel-1 soil moisture products were established in the Biebrza Wetlands in Northeastern Poland in May 2015 (Musial et al., 2016). One site is located across drained grassland and the second one across natural temporarily flooded marshland. They are located within 7 km distance, therefore weather conditions are similar but soil moisture regimes differ. Both sites are equipped with 9 soil moisture stations with 5 soil probes

each and a weather station. The sites are homogeneous regarding vegetation cover. The organic peat soils feature porosity values up to 82%. The sites are maintained by the Remote Sensing Centre of the Institute of Geodesy and Cartography (IGiK).

## A6 BNZ-LTER

The Bonanza Creek Long Term Ecological Research (BNZ-LTER) network consists of 12 stations located in the boreal forest near Fairbanks, Alaska (Van Cleve, 2015). Soil moisture measurements start around the year 2000 for each station. In addition

to soil moisture in several depths, observations are availble of soil temperature, air temperature, precipitation, snow depth and snow water equivalent.

## A7 CALABRIA

The Calabria network operates 5 TDR stations measuring volumetric soil moisture at 30, 60 and 90 cm depth. The stations were installed in 2000 by the "Centro Funzionale Decentrato" of the Calabria Region for hydrometeorological monitoring for

flood and landslide risk mitigation. For more information on the performance of the network see (Brocca et al., 2011).

## A8 CAMPANIA

The CAMPANIA network consisted of two stations located near the city of Naples, southern Italy. It was managed by the "Centro Funzionale per la Previsione Meteorologica e il Monitoraggio Meteo-Pluvio-Idrometrico e delle Frane". The ISMN contains data from the operational start in 2000 until the end of 2008. The data sets include soil moisture measured at a depth of

0.30 m, precipitation, and air temperature. For more information on the performance of the network, see (Brocca et al., 2011).

## A9 CARBOAFRICA

CARBOAFRICA is located outside El Obeid in Kordofan in Sudan and has been active since February 2002. It is operated by the Department of Physical Geography and Ecosystem Science of the Lund University, Sweden, in cooperation with the Agricultural Research Corporation (ARC) in El Obeid, Sudan. From 2007-2009, eddy covariance measurements were undertaken,

Which are available from FLUXNET. For more information, see Ardö (2013).

## A10   CHINA

This agricultural monitoring campaign from 1981 to 1999 includes 40 stations with soil moisture measurements until one meter depth and was hosted by the Institute of Geographical Sciences and Natural Resources Research of the Chinese Academy of Sciences in Beijing (Liu et al., 2001). The data set were transferred to the ISMN from the Global Soil Moisture Data Bank
(Robock et al., 2000) and contain measurements amde at the 8th, 18th and 28th of each month.

## A11   COSMOS

The Cosmic-ray Soil Moisture Observing System (COSMOS; Zreda et al., 2012) started in 2009 with a grant from the US National Science Foundation as a four-year project for demonstration of the then new technology of sensing soil moisture with cosmogenic neutrons (Zreda et al., 2008). On the completion of the project the network had 60 sites, most of them in the USA,
and a few in South America, Europe and Africa. The network produces hourly soil moisture data available in real time to all without restrictions. After the project funding ended in 2013, the network operations continued with the support by Quaesta Instruments, a private company. The current status is active, but the sensors are being relocated and repurposed.

## A12   CTP_SMTMN

CTP-SMTMN is a multiscale Soil Moisture and Temperature Monitoring Network on the central Tibetan Plateau (Yang et al.,
2013). The network, with an average elevation of 4650 m a.s.l., consists of 56 stations that measure soil moisture and soil freeze/thaw status at three spatial scales (100, 25, and 9 km). The terrain is relatively flat and covered by sparse and short grasses; the annual precipitation is about 400-50 0mm. The network has been in operation since 2010.

## A13   DAHRA

The Dahra field site is located in a typical low tree and shrub savanna environment in Senegal. To limit the uncertainty in the
comparison of remote sensing products and models, the site was selected to be flat with homogeneous vegetation cover within a radius of at least three km. The site is equipped with two towers: a meteorological tower with meteorological, hydrological and radiation sensors, and an eddy covariance flux tower. More info can be found at (**?**).

## A14   FLUXNET-AMERIFLUX

AMERIFLUX is the North-American contribution to the global FLUXNET. At this moment two sites close to Sacramento,
CA, i.e., Tonzi and Vaira Ranch, are distributed through the ISMN. Both stations provide soil moisture measurements at eight different depths down to 0.60 m. Additionally, soil temperature, air temperature, and precipitation are provided.

## A15   FMI

This distributed network of in situ measurement stations gathering information on soil moisture and soil temperature has been set up in recent years at the Finnish Meteorological Institute's (FMI) Sodankylä Arctic research station in Northern Finland.

Between 2010 and 2017, 16 stations were installed around Sodankylä and 3 further north at Saariselkä. Each station covers a vertical measuring profile and two additional horizontal measuring points. The vertical profiles have five sensors placed close to the station at the following depths: 5, 10, 20, 40, and 80 cm in mineral and semi-organic soils, and at 5, 10, 20, 30, 40 cm in organic soils. The two additional horizontal measuring points, at depths of 5 and 10 cm, have been installed approximately ten meters from the station in opposing directions to catch small-scale variations in top soil moisture. A more detailed description
is provided in (Ikonen et al., 2015, 2018).

### A16 FR_Aqui

The FR-AQUI network has been set up by INRAE in the Bordeaux–Aquitaine region (southwestern France) in 2012 (Wigneron et al., 2018; Al-Yaari et al., 2018a). Measurements taken at this five-station network (plus the nearby ICOS Bilos site), include soil moisture and temperature at various depths, and the height of the groundwater table. Four sites are installed in the Les
Landes forest, which is one of the largest coniferous forests in Europe, and one site is installed close to vineyards of the Bordeaux Graves region.

### A17 GROW

GROW gathered crowd sourced observations to assess the temporal and spatial consistency of various satellite-derived soil moisture products. 6,500 low cost sensors were deployed in 24 GROW places, in 13 Countries across Europe. A subset of 150
sensors were transferred to the ISMN (Zappa et al., 2019, 2020)) and contain measurements of Soil Moisture (%), Temperature (C) and metadata: Latitude and Longitude of the Sensor, Name of the Sensor, Date of Observation. Complete data set is licensed and available at https://doi.org/10.15132/10000156. More information on the quality of the data can be found in Xaver et al. (2020).

### A18 GTK

The GTK network is operated by the Geological Survey of Finland and contains 7 stations. The data are available from the years 2001 to 2012 but the availability varies per station.

### A19 HiWATER_EHWSN

The HiWATER EHWSN network is located in an irrigated farmland in the middle stream of the Heihe River Basin close to Gobi desert, China. It consists of short time series between April and September 2012 collected at 174 stations by the Cold
and Arid Regions Environmental and Engineering Research Institute (CAREERI) of the Chinese Academy of Science (Kang et al., 2014; Jin et al., 2014).

## A20   HOAL-SoilNet

The Hydrological Open Air Laboratory - Soil Network (HOAL-Soilnet) has been set up in Petzenkirchen, Austria, as part of a concerted effort to advance the understanding of water-related flow and transport processes in a 66 ha agricultural catchment (Blöschl et al., 2016). Soil moisture has been monitored since 2013 at about 30 locations at four depths at time intervals of 30 minutes using TDR sensors. Measurements are taken from permanent stations, located in grassland, forest or at field boundaries, as well as from stations that are temporarily installed in cropland (Vreugdenhil et al., 2013).

## A21   HOBE

HOBE is a hydrological observatory established in the western part of Denmark in the Skjern River catchment Jensen and Refsgaard (2018). Within the sub-catchment Ahlergaarde data have been collected from a network 30 soil moisture stations distributed within the sub-catchment according to respective fractions of classes representing mainly land cover and soil type (Bircher et al., 2012). At each station Decagon 5TE capacitance sensors have been installed at 2.5, 22.5 and 52.5 cm depths. The sensors are logged every 30 minutes.

## A22   HSC_SEOLMACHEON

HSC_SELMACHEON was a single station located in South Korea. Data were collected by the Hydrological Survey Center (HSC) and WRRSL and are available for the period August - September 2011.

## A23   HYDROL-Net Perugia

The HYDROL-Net Perugia network Morbidelli et al. (2014, 2017) consists of 1 station (called WEEF: Water Engineering Experimental Field) located near the city of Perugia, Central Italy. Measurements of soil moisture at 4 different depths (0.05, 0.15, 0.25 and 0.35 m), soil temperature at 2 depths (0.05 and 0.30 m), precipitation and air temperature from January 2010 to December 2013 are included in the ISMN. HYDROL-Net Perugia is operated by the Department of Civil and Environmental Engineering of the Perugia University.

## A24   ICN

The former Illinois Climate Network was operated by the Water and Atmospheric Resources Program of the Illinois State Water Survey and formerly integrated in the Global Soil Moisture Data Bank (Hollinger and Isard, 1994). Between 1983 and 2010, ICN covered 19 stations measuring soil moisture and soil temperature down to two metres depth, as well as precipitation.

## A25   IIT_KANPUR

The network IIT_KANPUR network consisted of a single station and was managed by Hydraulics and Water Resources Laboratory, Indian Institute of Technology Kanpur, India. Soil moisture measurements were made at four depths (10, 25, 50 and 80



cm) between from June 2011 to November 2012. The station was situated in the Ganga river basin, which is the largest river basin in India and the soil type at the observation site is clayey silt.

### A26 IMA_CAN1

The IMA_CAN1 network is operated by the Institute for Agricultural and Earthmoving Machines (IMAMOTER) of the Italian National Research Council (CNR), now part of STEMS-CNR. It is located in an experimental vineyard in Carpeneto, in
the "Alto Monferrato" hilly region, which is a valuable vine-growing and wine production area in the Piedmont Region in northwestern Italy. The monitored vineyards are part of the "Experimental Vine and Wine Centre of Agrion Foundation". The stations in the network provide measurements of soil moisture, precipitation, air temperature and humidity, hourly runoff and event soil losses. Hourly volumetric soil moisture was measured by 12 5TM sensors in the period 2011 - 2015, both in grassed and tilled vineyards, in correspondence of the track and no-track position, down and up the hill (Biddoccu et al., 2016; Capello
et al., 2019; Raffelli et al., 2018).

### A27 IOWA

The IOWA network was located in two catchments in the southwestern part of Iowa, USA. Soil moisture observations from 6 stations, until 2.6 meter depth, with an interval of twice a month during 1972 to 1994 (April to October) are included in the ISMN (Robock et al., 2000). This network was formerly included in the Global Soil Moisture Data Bank.

### A28 IPE

The Instituto Pirenaico de Ecologia (IPE) network runs two stations located in Aragón, north-eastern Spain. The stations have been collecting meteorological data with at least hourly time resolution (air and soil temperatures, soil moisture, relative humidity, radiation, wind speed) since 2008 in a Mediterranean oak forest (Agüero) and a semi-arid pine forest (Peñaflor). These measurements are related to dendrometer hourly records of changes in root and stem (Agüero; see (Alday et al., 2020))
or stem (Peñaflor) increment of the main tree species.

### A29 iRON

The interactive Roaring Fork Observation Network (iRON) is a series of 10 stations operated by the Aspen Global Change Institute spread across the elevations of the Roaring Fork Watershed, located in the Southern Rocky Mountains of the United States. This data set includes soil moisture at 5, 20, and 50cm, soil temperature at 20cm, and additional weather measurements
variable by station. Further information can be found in Osenga et al. (2019) or on agci.org/iron/about.

### A30 KHOREZM

The KHOREZM network in Uzbekistan is located between the Amudarya river and the border to Turkmenistan and was part of a project conducted by Uni Würzburg, Germany (Patrick Knöfel). Seven stations with soil moisture, soi temperature, air tem-



perature and surface temperature measurements from 2010 April to 2011 September are included in the ISMN. Although soil
moisture was not observed continuously, the measurements are still a valuable contribution since no other recent observations
are available in this region.

### A31   KIHS_CMC

The Korea Institute of Hydrological Survey has been running the Cheongmicheon (CMC) network since 2009 with annually
returning measurements from March till December. It comprises 56 TDR Buriable Waveguide soil moisture sensors at 18
stations located on an area the size of a football field. All stations have a sensor installed at 0.1 m and additional sensors at
varying depths, i.e., at 0.3, 0.4, 0.6 and 0.9 m.

### A32   KIHS_SMC

KIHS_SMC is operated by the Enivronmental and Remote Sensing Lab of the Korea Institute of Hydrological Survey. The 51
soil moisture sensors (depths from 0.1 to 0.6 m) are located on a mountain slope distributed over 19 stations in close proximity
to each other.

### A33   LAB-net

LAB-net was created as the first soil moisture network in Chile to support remote sensing research on drought and water
use conflicts (Mattar et al., 2016). Three stations measuring soil moisture, soil temperature, precipitation, and air temperature
between 2014 until 2017 over various land cover types have been included in the ISMN.

### A34   MAQU

The Maqu monitoring network ((Su et al., 2011; Dente et al., 2012) is situated on the north-eastern fringe of the Tibetan
Plateau, covering an area of approximately 40 km×80 km with the elevation varying from 3200 m to 4200 m above sea level.
The network have SMST profiles (5 cm, 10 cm, 20 cm, 40 cm, and 80 cm) measured with a 15-min interval. Soil moisture and
temperature data from 2008 to 2018 are included in the ISMN.

### A35   METEROBS

The METEROBS (MET European Research OBServations) network measured soil moisture at a single site between October
2011 October and May 2012. It was located in the Apennines mountains in the rural area of Benevento, Southern Italy. Soil
Moisture measurements from five layers until 0.5 meter depth have been included in the ISMN.

### A36   MOL-RAO

The MOL-RAO network is operated by the German Meteorological Service (DWD) and part of the operational measuring
program of the MOL-RAO (Meteorological Observatory Lindenberg – Richard-Aßmann-Observatory). The network is situated



in the north-east of Germany and consists of two stations. While the Station Falkenberg has a grass type vegetation, Kehrigk is situated in a pine forest (Beyrich and Adam, 2007). Volumetric soil moisture, soil temperature, air temperature and precipitation are provided in half hour resolution since 2003.

### A37 MONGOLIA

Soil moisture data sets for 44 stations were collected by the National Agency of Meterology, Hydrology and Environment Monitoring in Ulaanbaatar. All observations were taken using the gravimetric technique and initially provided as volumetric plant-available water [%]. Volumetric soil moisture [$m^3/m^3$] was calculated by first extracting texture properties of all sites from the Harmonized World Soil Database and subsequently calculating the wilting levels for all stations using the equations of Saxton and Rawls (2006). Soil moisture measurements are provided three times a month ( on the $8^th$, $18^th$ and $28^th$)from 1964 to 2002 during the warm period of the year, which runs from April until the end of October (e.g., Robock et al. (2000)).

### A38 MySMNet

The Malaysian Soil Moisture Network (MySMNet) is operational since 2014. It deploys seven stations (four on oil palm plantation, two on shrubland and one on orchard) that collect soil moisture at 5, 50, and 100 cm depth, soil temperature at 5 cm, air temperature, and relative humidity, all on an hourly basis. The soil moisture sensors used are the WaterScout SM100 (Kang et al., 2019).

### A39 ORACLE

The network ORACLE network includes 6 stations. The datasets reach back to the year 1985 and are available until 2013. ORACLE is a research observatory east of Paris to study of the "Grand Morin" and "Petit Morin" river catchments, particularly floods, low water periods, water quality, and the impact of human activities on the environment.

### A40 OZNET

OzNet was established in 2001 with 8 sites across the Murrumbidgee catachment and a cluster of a further 5 sites in the Adelong and Kyeamba catchments. This was further extended in 2003 to include a total of 11 sites at Kyeamba for a GRACE validation experiment and 13 sites at Yanco for SMOS pre-launch algorithm development and post-launch calibration-validation. The Yanco site was further extended to have clusters of stations across 9 km and 3 km grids in crop and grasslands areas for SMAP algorithm development, calibration, and validation (Smith et al., 2012).

### A41 PBO_H2O

The PBO_H2O network is a former near real time network of the ISMN hosted by the University of Colorado, USA. It consisted of 159 stations distributed in the west of the USA, Bahamas, Dominican Republic, Puerto Rico, Colombia, South Africa and Saudi Arabia. Soil moisture (measured using GPS reflections), precipitation, air temperature and snow depth from 2004 to





2017 is stored in the ISMN database (Larson et al., 2008). The network was discontinued in 2018 because of lacking financial support.

## A42 PTSMN

The Patitapu Soil Moisture Network (PTSMN) was established in 2016 on the hill country landscapes of the East Coast of
New Zealand's North Island. PTSMN was deployed to capture spatiotemporal soil moisture trends on various topographical positions distributed over a 13.8 km2 area. The network is composed of twenty multi-sensor probes that were calibrated to the site-specific soils (Hajdu et al., 2019). The sensors provide readings at four consecutive depths down to 0.43 m.

## A43 REMEDHUS

REMEDHUS is the University of Salamanca Soil Moisture Measurement Stations Network and was installed in March 1999,
with a set of old TDR stations and manual measurements. It is one of the first soil moisture networks in Europe. The network was automated and upgraded with capacitance probes in 2005. The REMEDHUS data available in the ISMN cover the period since the automation. REMEDHUS is located in an agricultural area in the central part of the Duero basin (Spain). The network currently has 20 stations that hourly measure soil moisture and soil temperature in the 0-5 cm layer (González-Zamora et al., 2018).

## A44 RISMA

The Real-time in situ Soil Monitoring for Agriculture (RISMA) network was established in 2011 by Agriculture and Agri-Food Canada at agricultural locations in Ontario, Manitoba, and Saskatchewan(Ojo et al., 2015; J., 2011; Canisius, 2011). There are currently 23 RISMA stations collecting hourly soil moisture/temperature data at depths to 1 – 1.5 metres in combination with meteorological data. Calibrated, quality controlled soil moisture and weather data are provided to the ISMN on an annual basis.

## A45 RSMN

The Romanian Soil Moisture Network (RSMN) consists of 19 stations homogeneously distributed over Romania. The network is managed by the Romanian National Meteorological Administration and is part of the ASSIMO project, which aims to create a framework for the evaluation of current and future satellite microwave-derived soil moisture products.

## A46 RUSWET-AGRO, RUSWET-GRASS, RUSWET-VALDAI

The three historical RUSWET networks were agricultural prediction campaigns conducted by the State Hydrological Institute of the Former Soviet Union within the area of present Russia. Measurements were taken from 1952 until 2002 and initially distributed through the Global Soil Moisture Data Bank. Altogether, the networks operated 337 sites at which soil moisture was measured three times per month via gravimetric sampling. At RUSWET-VALDAI also soil temperature, precipitation, and





air temperature were collected. RUSWET contributes both the northernmost (on Mc-Clintock Island) and the earliest (June 8th
1952) observations to the ISMN (Robock et al., 2000).

### A47  SASMAS

The Scaling and Assimilation of Soil Moisture and Streamflow (SASMAS) monitoring network commenced commissioning in
late 2002 with a total of 26 stations in operation by 2003 across a 6,500 km2 catchment (Rüdiger et al., 2007). Soil moisture is
observed up to a depth of 90 cm (where possible) at depth intervals of 0-30, 30-60, and 60-90 cm. The network was developed
as a nested catchment with three different types of scales (low-resolution across the entire catchment, medium density across
to smaller subcatchment, and very high density across a 175 ha single reach. This site also hosted the second NAFE campaign
for which additional data was collected.

### A48  SCAN

The Natural Resources Conservation Service (NRCS) operates the comprehensive, US-wide Soil Climate Analysis Network.
SCAN supports natural resource assessments and conservation activities through its network of automated climate monitoring
and data collection sites. SCAN focuses primarily on agricultural areas of the U.S., Puerto Rico and the Virgin Islands. The
network consists of 216 stations located across the United States and reports soil moisture, soil temperature, precipitation,
temperature and other climatic variables hourly. Soil sensors are situated at 5, 10, 20, 50 and 100 cm depth (Schaefer et al.,
2007).

### A49  SKKU

The SKKU network was located at the evenly and moderately vegetated botanical garden in South Korea. It was operated
by Sungkyunkwan University from 2014 to 2016 as part of a project for evaluating Cosmic-Ray Neutron Probe (CRNP) soil
moisture (Nguyen et al., 2017). The network consisted of 10 stations, and soil moisture measurements were taken from 4
different depths (i.e., 10, 20, 30, and 40 cm).

### A50  SMOSMANIA

The SMOSMANIA network was installed in southern France by Météo-France, the French national meteorological ser-
vice, in order to monitor in situ soil moisture and soil temperature in contrasting soil and climatic conditions at operational
automatic weather stations (Calvet et al., 2007). The SMOSMANIA network is composed of 21 stations forming an At-
lantic–Mediterranean transect over a large variety of mineral soils ranging from sand to clay and silt loam (Calvet et al., 2016).

### A51  SNOTEL

NRCS installs, operates, and maintains an extensive, automated data collection network called SNOTEL (short for Snow
Telemetry) (Leavesley et al., 2008; Leavesley, 2010). SNOTEL is part of the Snow Survey and Water Supply Forecasting

(SSWSF) Program and is designed to collect snowpack and related climatic data in the western U.S. and Alaska. The Program operates under technical guidance from the NRCS National Water and Climate Center (NWCC). With the majority of the water
supply in the West arriving in the form of snow, data on snowpack provide critical information to decision-makers and water managers. SNOTEL currently consists of a network of over 860 automated SNOTEL stations of which 463 stations have soil moisture and soil temperature sensors.

### A52  SOILSCAPE

The Soil moisture Sensing Controller And oPtimal Estimator (SoilSCAPE) is a wireless soil moisture sensor network for
measurements of surface-to-root-zone profiles of soil moisture (Moghaddam et al., 2011; MOGHADDAM et al., 2016; Shuman et al., 2010). It is designed for long-term, ultra-efficient, unattended field operations. Several networks are deployed in the US, including in the states of California, Arizona, Colorado, Alaska, and New York. Additional deployments are planned in New Mexico, US, and in New Zealand. The SoilSCAPE architecture includes a local coordinator (LC) and multiple end devices (EDs). The LC is the central command center of the network, receiving data from the several soil moisture and temperature
sensors connected to each ED. The sensors include 5TM and Teros-12, depending on how recently they were deployed.

### A53  SWEX_POLAND

The oSil Water and Energy exchange - Poland (SWEX_POLAND) network was operated between 2000 and 2013 by the Institutes of Agrophysics PAS in Lublin. The network consisted of 6 stations located in the wetlands of Poleski Park Krajobrazowy to support SMOS product calibration and validaton. Soil moisture and temperature measurements were taken down to 1 meter
depth, along with precipitation observations (Marczewski et al., 2010).

### A54  SW-WHU

The SW-WHU network was hosted by Wuhan University (Chen et al., 2015a, b). SW-WHU is a high-density network, with nearly 100 soil moisture and temperature sensors within 1 km2. It adopts an NB-IOT technique for data transmission at low power consumption. Therefore, the data of SW-WHU is particularly valuable for soil moisture validation, monitoring, and
application at a very high spatio-temporal resolution (Zhang et al., 2018b).

### A55  TAHMO

The Trans-African Hydro-Meteorological Observatory (TAHMO; tahmo.org) presently runs a network of over 600 meteorological weather stations in more than twenty African countries. The stations measure all standard weather parameters, such as barometric pressure, wind speed, rainfall, and radiation. Each station also has five open ports that can be used for additional
sensors, such as soil moisture sensors. Presently, 70 stations have soil moisture sensors, some of them at several depths. Ideally, many stations will be upgraded by soil moisture sensors in the near future.





## A56   TERENO

TERENO consists of four terrestrial observatories that represent typical landscapes in Germany and Central Europe and are considered to be highly vulnerable to the effects of global and climate change (Zacharias et al., 2011; Bogena et al., 2012).
TERENO combines observations with comprehensive large-scale experiments, integrated modeling, remote sensing and novel measurement technologies to increase our understanding of the functioning and feedbacks of terrestrial ecosystems (Bogena, 2016). The long-term observation platform of TERENO is composed of various measurement systems, including networks of climate and lysimeter stations, eddy-covariance towers, and networks of soil, surface water and groundwater sensors. Almost all online measurements are freely accessible via the TERENO data portal (http://www.tereno.net/ddp/).

## A57   UDC_SMOS

The former German UDC_SMOS network was hosted by the University of Munich, Department of Geography in cooperation with the Bavarian Sate Research Center for Agriculture and funded by the German Aerospace Centre (DLR). It was located in grassland in the Bavarian region around Munich city, as an official European SMOS cal/val test site. Eleven stations provided soil moisture data from 2007 until 2011 until 40 cmr depth measured by several types of sensors (Loew et al., 2009; Schlenz
et al., 2012).

## A58   UMBRIA

This soil moisture monitoring network in the north of the Umbria region, in the upper Tiber River basin operates three stations in real-time: Torre dell'Olmo, Petrelle, Cerbara. Each station measures at 10 cm, 20 cm, and 40 cm depth. Additional stations of the network have not been operational since 2015 due to a lack of resources for their maintenance. Provided financial resources
become available, the stations that are no longer functioning will be restored.

## A59   UMSUOL

UMSUOL ("Umidita del Suolo") is a one-station network located close to Bologna, northern Italy. Soil moisture measurements at seven different depths are provided by the Agenzia Regionale Prevenione Ambiente (ARPA). The ISMN contains data from the years 2009 and 2010.

## A60   USCRN

The U.S. Climate Reference Network (USCRN) contains 114 stations sparsely distributed across the contiguous United States. Each station has three sets of soil moisture/temperature probes at 5 depths: 5, 10, 20, 50, and 100 cm in addition to air and surface temperature, precipitation, relative humidity, and solar radiation (Bell et al., 2013). The stations were installed between 2009 and 2011 and are still in operation in 2021. The stations are maintained by the National Oceanic and Atmospheric
Administration Atmospheric Turbulence and Diffusion Division (NOAA ATDD) and their data are maintained by NOAA National Centers for Environmental Information (NOAA NCEI).

## A61 USDA-ARS

The United States Department of Agriculture-Agricultural Research Service (USDA-ARS) operates a number of Long Term Agroecosystem Research (LTAR) sites, some of which have spatial coverage of soil moisture and soil temperature. As exper-
imental sites, the locations and configuration of the stations can change depending on the current scientific questions being addressed. A description of the sites can be found in Jackson et al. (2011).

## A62 VDS

The VDS network is run by VanderSat, a dutch company specialised in providing global satellite-observed data and services over land. The VDS network consists of four stations located near the city of Bago in Myanmar. The network was installed to
validate satellite soil moisture products in the tropics. The network has two measurement periods, one from June 2017 to July 2018 and one from March 2020 onwards.

## A63 VAS

The "Valencia Anchor Station" networis operated by the Climatology from Satellites Group and Jucar River Basin Authority of the University of Valencia, Spain. The network is located in Spain and consists 3 stations. The datasets are available for the
years 2010 and 2011.

## A64 WEGENERNET

The WegenerNet, located in the foreland of the southeastern Austrian Alps, is a long-term weather and climate monitoring facility comprising 155 hydrometeorological stations in a dense grid with one station every about 2 km$^2$ Kirchengast et al. (2014); Fuchsberger et al. (2020). Together with a range of meteorological variables such as temperature, humidity, precipita-
tion and wind, it also measures soil moisture and temperature at 12 stations. These variables are measured at 0.2-0.3 m depth in diverse soil types representative for the region.

## A65 WSMN

The Wales Soil Moisture Network was founded in July 2011. It consists of a total of 9 monitoring sites located in Mid-Wales representing a range of conditions typical of the Welsh environment, with climate ranging from oceanic to temperate and the
most typical land use/cover types. The data set acquired in the network are composed of 0-5 (or 0-10) cm soil moisture, soil temperature, precipitation as well as other ancillary data (Petropoulos and McCalmont, 2017).





## Appendix B:  Network abbreviations

| Acronym/ definition | Full name |
|---|---|
| AACES | Australian Airborne Cal/val Experiment for SMOS |
| AMMA-CATCH | African Monsoon Multidisciplinary Analysis – Coupling the Tropical Atmosphere and the Hydrological Cycle |
| ARM | Atmospheric Radiation Measurement Climate Research Facility |
| AWDN | Automated Weather Data Network (name of in situ network in the USA) |
| BIEBRZA_S-1 | Biebrza Sentinel-1 Supersite |
| BNZ-LTER | Bonanza Creek Long-Term Ecological Research |
| CALABRIA | Name of in situ network in Italy |
| CAMPANIA | Name of in situ network in Italy |
| CARBOAFRICA | Name of in situ network in Sudan |
| CHINA | Name of in situ network in China |
| COSMOS | Cosmic-ray Soil Moisture Observing System |
| CTP_SMTMN | Central Tibetan Plateau Soil Moisture and Temperature Monitoring Network |
| DAHRA | Name of in situ network in Senegal |
| ECMWF | European Centre for Medium-Range Weather Forecasts |
| FLUXNET_AMERIFLUX | Name of in situ network in USA |
| FMI | Finnish Meteorological Institute (name of in situ network in Finland) |
| FR_Aqui | Name of in situ network in France |
| GDPR | General Data Protection Regulation |
| GROW | GROW citizen science project (name of in situ network based in the UK but distributed through out Europe) |
| GTK | Geological Survey of Finland (name of in situ network in Finland) |
| HiWATER_EHWSN | Heihe Watershed Allied Telemetry Research Wireless Sensor Network (in China) |
| HOBE | Hydrological Oberservatory in Denmark |
| HSC_SEOLMACHEON | Hydrological Survey Center (HSC) in –situ network in South Korea |
| HWSD | Harmonized World Soil Database |
| HYDROL-NET_PERUGIA | Name of in situ network in Italy |
| ICN | Illinois Climate Network (name of in situ network in USA) |
| IIT_KANPUR | Indian Institute of Technology Kanpur (name of in situ network in India) |
| IGiK | Institute of Geodesy and Cartography |

*Table continued on next page*





| Acronym/ definition | Full name |
|---|---|
| IMA_CAN1 | STEMS-CNR in situ soil monitoring at the Tenuta Cannona Experimental Vine and Wine Centre of Agrion Foundation |
| IOWA | Name of in situ network in USA |
| IPE | Instituto Pirenaico de Ecologia (Organisation name and the name of in situ network in Spain) |
| iRON | Roaring Fork Observation Network |
| ISMN | International Soil Moisture Network |
| KHOREZM | Name of in situ network in Uzbekistan |
| KIHS_CMC | Name of in situ network in South Korea |
| KIHS_SMC | Name of in situ network in South Korea |
| LAB-net | Laboratory for Analysis of the Biosphere Network (name of in situ network in Chile) |
| MAQU | Name of in situ network in China |
| METEROBS | Met European Research Observatory (name of in situ network in Italy) |
| MOL-RAO | Lindenberg Meteorological Observatory, Richard-Aßmann Observatory |
| MONGOLIA | Name of in situ network in Mongolia |
| MySMNet | Name of in situ network in Malaysia |
| NRT | Near Real Time |
| ORACLE | Name of in situ network in France |
| OZNET | Name of in situ network in Australia |
| PBO_H2O | Plate Boundary Observatory (Name of in situ network in USA) |
| REMEDHUS | University of Salamanca Soil Moisture Measurement Stations Network |
| RISMA | Real-time in situ Soil Monitoring for Agriculture Network in Canada |
| RSMN | Romanian Soil Moisture Network |
| RUSWET-AGRO | Name of in situ network in Former Soviet Union |
| RUSWET-GRASS | Name of in situ network in Former Soviet Union |
| RUSWET-VALDAI | Name of in situ network in Former Soviet Union |
| SASMAS | Scaling and Assimilaton of Soil Moisture and Streamflow |
| SCAN | Soil Climate Analysis Network |
| SWATS | Surface Energy Balance System, sensor network of ARM |
| SKKU | Name of in situ network in South Korea |
| SMOS | Soil Moisture and Ocean Salinity |
| SMOSMANIA | Soil Moisture Observing System – Meteorological Automatic Network Integrated Application |
| SNOTEL | SNOwpack TELemetry |

*Table continued on next page*



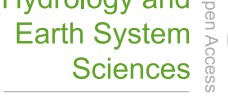

| Acronym/ definition | Full name |
|---|---|
| SOILSCAPE | Soil Moisture Sensing Controller and Optimal Estimator Network in the USA |
| STAMP | Soil Temperature and Moisture Profile, sensor network of ARM |
| SWATS | Soil Water and Temperature System, sensor network of ARM |
| SWEX_POLAND | Soil Water Experiment Poland (name of in situ network in Poland) |
| SW-WHU | Name of in situ network in Poland |
| TAHMO | Trans-African Hydro-Meteorological Observatory (name of in situ network in Sahel Zone in Africa) |
| TERENO | Terrestrial Environmental Observatories (name of in situ network in Germany) |
| UDC_SMOS | Upper Danube Catchment SMOS (name of in situ network in Germany) |
| UMBRIA | Name of in situ network in Italy |
| UMSUOL | Umidita del Suolo (name of in situ network in Italy) |
| USCRN | US Climate Reference Network |
| USDA_ARS | Research Watersheds Supporting Soil Moisture Remote Sensing Network in USA |
| UTC | Universal Time Coordinated |
| VAS | Valencia Anchor Station |
| VDS | Vandersat, in situ network in Myanmar |
| WEGENERNET | Name of in situ network in Austria |
| WSMN | Wales Soil Moisture Network in the UK |





## Appendix C: Detailed system overview



**Figure C1.** Detailed system overview of the ISMN (Status January 2021).



## Appendix D:  Soil moisture sensor availability per soil depth interval

**Table D1.** Overview of data availability per depth layers showing the total number of individual sensor brands, data sets and stations integrated in the ISMN database. The calculations have been made in respect to horizontally and vertically placed sensors for each depth layer.

| depth from - to [m] | sensors | data sets | stations |
| --- | --- | --- | --- |
| 0.00 - 0.05 | 131 | 2116 | 1696 |
| 0.05 - 0.10 | 123 | 2482 | 1459 |
| 0.10 - 0.20 | 138 | 2690 | 1523 |
| 0.20 - 0.30 | 104 | 1899 | 1463 |
| 0.30 - 0.40 | 43 | 562 | 483 |
| 0.40 - 0.50 | 73 | 1463 | 1207 |
| 0.50 - 0.60 | 65 | 1389 | 1125 |
| 0.60 - 0.70 | 25 | 210 | 185 |
| 0.70 - 0.80 | 20 | 113 | 106 |
| 0.80 - 0.90 | 24 | 183 | 159 |
| 0.90 - 1.00 | 25 | 346 | 302 |
| 1.00 - 1.10 | 42 | 571 | 450 |
| 1.10 - 1.20 | 5 | 7 | 5 |
| 1.20 - 1.30 | 7 | 44 | 24 |
| 1.30 - 1.40 | 4 | 8 | 6 |
| 1.40 - 1.50 | 8 | 13 | 9 |
| 1.50 - 1.60 | 12 | 21 | 15 |
| 1.60 - 1.70 | 0 | 0 | 0 |
| 1.70 - 1.80 | 3 | 35 | 18 |
| 1.80 - 1.90 | 1 | 1 | 1 |
| 1.90 - 2.00 | 8 | 20 | 20 |
| 2.00 - 2.10 | 13 | 10 | 8 |





**Appendix E:  Metadata structure**

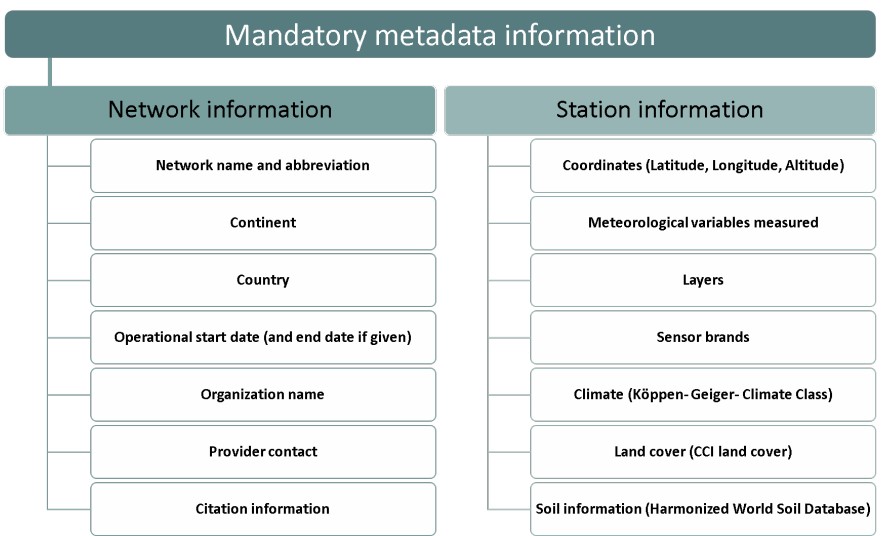

**Figure E1.** Overview of all mandatory metadata information available for each station within the ISMN database.



## Appendix F: Scientific studies using ISMN data for satellite and model product evaluation and development.

**Table F1.** Overview of scientific studies using ISMN data for satellite product evaluation and development.

| Sensor | Publications |
|---|---|
| SMOS | Albergel et al. (2012c), Mecklenburg et al. (2012), Parrens et al. (2012), Sanchez et al. (2012), Schlenz et al. (2012), Wanders et al. (2012), Pellarin et al. (2013), Fascetti et al. (2014), Petropoulos et al. (2014), Pierdicca et al. (2014), Zeng et al. (2014), Rodriguez-Fernandez et al. (2015), Gruber et al. (2015), Hottenstein et al. (2015), Kornelsen and Coulibaly (2015), Lee and Im (2015), Parinussa et al. (2015), Pierdicca et al. (2015a), Pierdicca et al. (2015b), Scaini et al. (2015), González-Zamora et al. (2015), Fascetti et al. (2016), Fascetti et al. (2016), Kędzior and Zawadzki (2016), Kerr et al. (2016), Pablos et al. (2016), Piles et al. (2016), Rautiainen et al. (2016), Scholze et al. (2016), van der Schalie et al. (2016), González-Zamora et al. (2016), Ángel González-Zamora et al. (2016), Al-Yaari et al. (2017), Cui et al. (2017), Fascetti et al. (2017), Fernandez-Moran et al. (2017b), Fernandez-Moran et al. (2017a), Karthikeyan et al. (2017), Lievens et al. (2017), Martens et al. (2017), Mohanty et al. (2017), Montzka et al. (2017), Muñoz Sabater et al. (2017), Pan et al. (2017), Peng et al. (2017), Pierdicca et al. (2017), Rains et al. (2017), van der Schalie et al. (2017), Yao et al. (2017), Al-Yaari et al. (2018b), Ebrahimi et al. (2018), Ebrahimi-Khusfi et al. (2018), Lee (2018), Liu et al. (2018), Pablos et al. (2018), Tagesson et al. (2018), Tian (2018), van der Schalie et al. (2018), Wu et al. (2018), Al-Yaari et al. (2019b), Al-Yaari et al. (2019a), Blyverket et al. (2019), Blyverket (2019), de Rosnay et al. (2019), **?**, Ma et al. (2019), Piles et al. (2019), Rodriguez-Fernandez et al. (2019), Tian et al. (2019b), González-Zamora et al. (2019), Beck et al. (2020), Chen et al. (2020), Helgert and Khodayar (2020), Herbert et al. (2020), Link et al. (2020), Moreno-Martínez et al. (2020), Portal et al. (2020), Sadri et al. (2020),Dong et al. (2020),Li et al. (2020c),Wigneron et al. (2021) |
| ASCAT | Brocca et al. (2011), Liu et al. (2011), Albergel et al. (2012c), Liu et al. (2012b), Parrens et al. (2012), Wanders et al. (2012), Albergel et al. (2013b), Pellarin et al. (2013), Zwieback et al. (2013), Fascetti et al. (2014), Gruber et al. (2014), Parinussa et al. (2014b), Paulik et al. (2014), Pierdicca et al. (2014), Gruber et al. (2015), Parinussa et al. (2015), Pierdicca et al. (2015a), Pierdicca et al. (2015b), Su et al. (2015), Zwieback et al. (2015), Fascetti et al. (2016), Fascetti et al. (2016), Zwieback et al. (2016), Fascetti et al. (2017), Gruber et al. (2017), Karthikeyan et al. (2017), Kolassa et al. (2017), Lievens et al. (2017), Massari et al. (2017), Mohanty et al. (2017), Montzka et al. (2017), Pierdicca et al. (2017), Sure et al. (2017), Al-Yaari et al. (2018b), Bauer-Marschallinger et al. (2018), Kim et al. (2018), Afshar et al. (2019), Al-Yaari et al. (2019b), Fairbairn et al. (2019), Gruber et al. (2019a), Beck et al. (2020), Chen et al. (2020), Link et al. (2020), Moreno-Martínez et al. (2020), Zappa et al. (2020) |
| AMSR-E | Brocca et al. (2011), Liu et al. (2011), Liu et al. (2012b), Parinussa et al. (2012), Wanders et al. (2012), Albergel et al. (2013b), Dente et al. (2013), Dorigo et al. (2013), Zhao and Li (2013), de Jeu et al. (2014), Parinussa et al. (2014b), Zeng et al. (2014), Coopersmith et al. (2015), Parinussa et al. (2015), Su et al. (2015), Zhao and Li (2015), Al-Yaari et al. (2016), Du et al. (2016), Han et al. (2016), Parinussa et al. (2016), Santi et al. (2016), Zhao et al. (2016), Gruber et al. (2017), Han et al. (2017), Karthikeyan et al. (2017), Kolassa et al. (2017), Massari et al. (2017), Mohanty et al. (2017), Su et al. (2017), Sure et al. (2017), Tobin et al. (2017), van der Schalie et al. (2017), WANG Rui (2017), Yao et al. (2017), Lei et al. (2018), Liu et al. (2018), van der Schalie et al. (2018), Afshar et al. (2019), Hu et al. (2019), Xie et al. (2019), Chen et al. (2020), Deng et al. (2020c), Hagan et al. (2020),Dong et al. (2020) |
| SMAP | Fascetti et al. (2016), Zeng et al. (2016), Al-Yaari et al. (2017), Cui et al. (2017), Fascetti et al. (2017), Karthikeyan et al. (2017), Mohanty et al. (2017), Montzka et al. (2017), Pierdicca et al. (2017), Alemohammad et al. (2018), Ebrahimi et al. (2018), Ebrahimi-Khusfi et al. (2018), Kim et al. (2018), Kim and Lakshmi (2018), Kolassa et al. (2018), Pablos et al. (2018), Santi et al. (2018), Xu et al. (2018), Zhao et al. (2018), Al-Yaari et al. (2019b), Bai et al. (2019), Blyverket et al. (2019), Blyverket (2019), Ebtehaj and Bras (2019), Fang et al. (2019), Ma et al. (2019), Park et al. (2019), Zhang et al. (2019b), Beck et al. (2020), Chen et al. (2020), Fang et al. (2020), Gao et al. (2020), Gao et al. (2020), Link et al. (2020), Park et al. (2020), Portal et al. (2020), Sadri et al. (2020), Suman et al. (2020), Sun et al. (2020), Zappa et al. (2020) |



**Table F1 continued from previous page**

| Sensor | Publications |
|---|---|
| ESA CCI | Gruber et al. (2013), Enenkel et al. (2016), Su et al. (2016), Cui et al. (2017), Dorigo et al. (2017), Karthikeyan et al. (2017), Martens et al. (2017), Tobin et al. (2017), Al-Yaari et al. (2018b), Al-Yaari et al. (2019b), Al-Yaari et al. (2019a), Blyverket et al. (2019), Blyverket (2019), Gruber et al. (2019b), Guevara and Vargas (2019), Ma et al. (2019), González-Zamora et al. (2019), Zhu et al. (2019), Beck et al. (2020), Deng et al. (2020c), Kovačević et al. (2020), Zappa et al. (2020), Dorigo et al. (2015), Pratola et al. (2015), Nicolai-Shaw et al. (2015b), An et al. (2016), Liu et al. (2018), Liu et al. (2020),Chen et al. (2016) |
| AMSR2 | Kim et al. (2015a), Kim et al. (2015b), Parinussa et al. (2015), Kim et al. (2016), Wu et al. (2016), Anoop et al. (2017), Cui et al. (2017), Karthikeyan et al. (2017), Montzka et al. (2017), Yao et al. (2017), Fang et al. (2018), Kim et al. (2018), Liu et al. (2018), Santi et al. (2018), Hu et al. (2019), Ma et al. (2019), Beck et al. (2020), Chen et al. (2020), Hagan et al. (2020), Link et al. (2020), Moreno-Martínez et al. (2020), Anoop et al. (2017) |
| MODIS | Liu et al. (2014), Boussetta et al. (2015), Zhao and Li (2015), Pablos et al. (2016), Przeździecki et al. (2017), Alemohammad et al. (2018), Gumbricht (2018), Pablos et al. (2018), Kumar et al. (2019b), Park et al. (2019), Han et al. (2020) |
| Sentinel-1 | Paloscia et al. (2013), Liu et al. (2017), Bao et al. (2018), Bauer-Marschallinger et al. (2018), Dabrowska-Zielinska et al. (2018), Bauer-Marschallinger et al. (2019), Greifeneder et al. (2019), Rodionova (2019b), Rodionova (2019a), Wang et al. (2019a), Foucras et al. (2020), Han et al. (2020), Ma et al. (2020), Zappa et al. (2020) |
| SEVIRI | Zhao and Li (2013), Liu et al. (2014), Leng et al. (2015),Leng et al. (2016), Piles et al. (2016), Leng et al. (2017), Tagesson et al. (2018), Ghilain et al. (2019) |
| FY-3B | Parinussa et al. (2014a), Cui et al. (2017), Parinussa et al. (2018), Sheng et al. (2019), Chen et al. (2020), Hagan et al. (2020) |
| TMI | Liu et al. (2012b), Albergel et al. (2013b), Gruber et al. (2015), Karthikeyan et al. (2017), Chen et al. (2020), Hagan et al. (2020) |
| ERS | Liu et al. (2012b), Albergel et al. (2013b), Dente et al. (2013), Kolassa et al. (2013), Karthikeyan et al. (2017) |
| GNSS-IR (CYGNSS) | Zhang et al. (2018a), Kim and Lakshmi (2018), Xu et al. (2018), Eroglu et al. (2019), Chew and Small (2020), Senyurek et al. (2020) |
| SSM/I | Liu et al. (2012b), Albergel et al. (2013b), Kolassa et al. (2013), Karthikeyan et al. (2017), Hagan et al. (2020) |
| WindSat | Parinussa et al. (2012), Karthikeyan et al. (2017), Chen et al. (2020), Hagan et al. (2020) |
| ERS-2 | Reimer et al. (2012), Pierdicca et al. (2014) |
| Landsat | Zhao et al. (2017), Bao et al. (2018), Pradhan (2019) |
| GRACE | Khaki et al. (2019), Sadeghi et al. (2020) |
| Aquarius/SAC-D | González-Zamora et al. (2016) |
| GF-1 WFV | Xing et al. (2017) |
| HY-2 | Zhao et al. (2014) |
| MeMo | Beck et al. (2020) |
| SMmodel | Mimeau et al. (2020) |





**Table F2.** Overview of scientific studies using ISMN data for model evaluation and development.

| Model | Publications |
| --- | --- |
| ERA-Interim, ERA-Interim\Land | Albergel et al. (2012b), Albergel et al. (2012a),Albergel et al. (2013a), Albergel et al. (2013b), Gruber et al. (2013),Pierdicca et al. (2015a), Zwieback et al. (2013), Angevine et al. (2014), Balsamo et al. (2015),Cammalleri et al. (2015), Zwieback et al. (2015), Kim et al. (2016), Deng et al. (2020c), Deng et al. (2020b), Li et al. (2020a), Pierdicca et al. (2015b), Nicolai-Shaw et al. (2015b), Beck et al. (2020) |
| ECMWF IFS | Albergel et al. (2012c), Mecklenburg et al. (2012), Pellarin et al. (2013), Albergel et al. (2015), Rodriguez-Fernandez et al. (2015) |
| ERA-40 | Liu et al. (2012a) |
| ERA5 | Li et al. (2020a) |
| H-TESSEL | Albergel et al. (2010), Albergel et al. (2015), Dirmeyer et al. (2016) |
| ISBA | Albergel et al. (2010), Parrens et al. (2012), Barbu et al. (2014) |
| GLDAS Noah | Liu et al. (2011), Dorigo et al. (2013), Angevine et al. (2014), Bi et al. (2016), Dirmeyer et al. (2016), Zawadzki and Kędzior (2016), Pan et al. (2017), Lee (2018), McDonough et al. (2018), Mishra et al. (2018), Afshar et al. (2019), Kędzior and Zawadzki (2016), Beck et al. (2020), Deng et al. (2020c), Solander et al. (2020) |
| GLDAS Mosaic | Bi et al. (2016), Solander et al. (2020) |
| MERRA, MERRA\Land, MERRA-2 | Albergel et al. (2013b), Pellarin et al. (2013), Kim et al. (2016), Reichle et al. (2017), Draper and Reichle (2019), Wang et al. (2019c), Li et al. (2020a) |
| CLM | Cammalleri et al. (2015), Bi et al. (2016), Dirmeyer et al. (2016), Mishra et al. (2018), Zhao and Yang (2018) |
| VIC | Mishra et al. (2014), Bi et al. (2016), Mishra et al. (2018), Beck et al. (2020), Solander et al. (2020) |
| GLEAM | Lievens et al. (2017), Martens et al. (2017), Martens et al. (2018) |
| LDAS | Fairbairn et al. (2015), Albergel et al. (2018), Xia et al. (2019), Albergel et al. (2020) |
| BEACH | Van doninck et al. (2012) |
| CAS-LSM | Wang et al. (2019b) |
| CFSR | Li et al. (2020a) |
| CLSM | Dong et al. (2020) |
| CPC | Liu et al. (2012a) |
| CSSP | Ji et al. (2017) |
| ESSMRA | Naz et al. (2020) |
| GSFC | Dirmeyer et al. (2016) |
| HBV | Beck et al. (2020) |
| JRA-5 | Li et al. (2020a) |
| LISFLOOD | Cammalleri et al. (2015) |
| NLDAS Noah | Dong et al. (2020) |
| NLDAS Mosaic | Dong et al. (2020) |
| NCEP/NCAR | Pierdicca et al. (2015b) |
| Noah-MP | Dong et al. (2020) |
| SSA | Fairbairn et al. (2019) |
| SSMP | Pal and Maity (2019) |
| WCDA, SCDA, OPNL | Lin and Pu (2019) |
| WRF, WRF3.3-CLM4crop | Angevine et al. (2014), Lu et al. (2015) |
| ORCHIDEE-MICT | Yin et al. (2018) |





## Appendix G: Effect of quality flagging on validation of ERA5 and ESA CCI soil moisture

(a) ERA5 absolute values

(b) ERA5 anomalies

(c) ESA CCI absolute values

(d) ESA CCI anomalies

**Figure G1.** Scatterplots of R Spearman between ISMN and ERA5 and ISMN and ESA CCI v05.2 combined for absolute values and anomalies. A 35-day moving average window was used to calculate anomalies. Each dot represents a single time series.





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
