# Peer review of "The International Soil Moisture Network: serving Earth system science for over a decade"

_Hydrology and Earth System Sciences, 2021_

## Author Comment (AC1)

**1 Reply to Jianzhi Dong, 03 Mar 2021**

We thank Jianzhi Dong for his positive view on the ISMN and the manuscript. Our suggestions to improve the manuscript are given in blue font.

Here is my review for "The International Soil Moisture Network: serving Earth system science for over a decade" by Wouter Dorigo et al. This manuscript provides a comprehensive review of ISMN. ISMN is an important dataset for a range of research topics, and this manuscript is an excellent tutorial of the dataset. Therefore, I would suggest accepting the manuscript after some minor revisions:

1. Figure 2: It is kind of difficult to quickly figure out which part of the world is still having active sensors. Therefore, I would suggest the author to visualizing the information on a global map.

The aim of this plot is to show the period of measurement of the individual networks, therefore we would like to keep the bar plot format. However, we agree that it would be informative for the user to see at a single glance which stations are active and which are not. Therefore, we propose to add a map (either in the main text or supplement, showing the stations stratified into networks that are updated in near-real-time, networks that are updated regularly (up till once per year), networks that are being updated irregularly, and networks that are no longer actively producing data (see revised figure below)..

2. Section 2.2: It is unclear if the sampling volume differences of different sensors are considered in ISMN.

The ISMN does not harmomise for differences in the sampling volume between sensors. To state this more clearly we propose to add the following clarification following line 126: "Note, that even if all measurements are harmonised in terms of units, differences in sampling volumes related to the sensor design and installation are not accounted for."

3. Figure 6: Please clarify if the calculated representativeness error includes the multiplicative and additive biases due to scale mismatch.

The errors computed with the triple collocation analysis are a combination of random measurement errors and random errors related to differences in scale. Systematic errors related to scale differences, i.e., multiplicative and additive biases, are being removed by scaling the land surface model and Earth observation datasets to the in situ data prior to performing the triple collocation analysis. To clarify this, we propose to add the following sentence after line 257: "Systematic differences between the datasets, i.e., multiplicative and additive biases, are being removed by scaling the land surface model and Earth observation datasets to the in situ data prior to the triple collocation analysis." 4. Line 325: a reference is missing

We will add the missing reference: "Bayat, B., Camacho, F., Nickeson, J., Cosh, M., Bolten, J., Vereecken, H., and Montzka, C.: Toward operational validation systems for globalsatellite-based terrestrial essential climate variables, International Journal of Applied Earth Observation and Geoinformation, 95, 102 240,https://doi.org/10.1016/j.jag.2020.102240, 2021."

5. Section 5.1.2: Dong et al., 2020 shows that, when sampled across a range of (sparse) sites representativeness error is random in nature. Therefore, when averaging across multiple sites, sparse sites can still accurately capture the \*relative\* accuracy of different soil moisture products.

We propose to add the following sentence at the end of this section: "Even though differences in spatial representativeness between ground and satellite measurements impact the evaluation metrics, single stations are still a valuable source for assessing the relative skill of soil moisture products with a similar footprint (Dong et al., 2020)".

6. Line 510: Please also include Active and Passive Distributed Temperature Sensing (DTS) works here, which is also aimed to measure soil moisture with high spatial resolutions at low costs and perform soil moisture scaling analysis.

Our intention of this section is not to be complete but to provide an example. This is also why we start line 510 with "For instance". In the preceding paragraph we already mention that a "rich variety of low-cost sensors based on different measurement principles has been developed". Hence, we propose not to go into more depth here.

Figure 1: Locations of ISMN sites classified either as near-real-time, as updated regularly (up till once per year), as being updated irregularly, and as stations that are no longer actively producing data

---

## Author Comment (AC2)

**1 Reply to Jan Friesen, 08 Mar 2021**

We thank Jan Friesen for underlining the relevance of ISMN and the manuscript for the scientific community. Our suggestions to improve the manuscript are given in blue font.

The manuscript 'The International Soil Moisture Network: serving Earth system science for over a decade' by Dorigo et al. presents the ISMN, its data providers and tools for quality control and data harmonization. The review also highlights the large diversity of data and its challenges but also the large number of studies that used and benefits from this unique data collection.

Besides a concise description of the ISMN data and the ISMN services in terms of harmonization, where possible, I also see this review as a useful reminder and call for networks and scientists to further use and support this initiative.

Detailed comments:

- Line 11: Please, state the number or percentage of data sets that are continuously/regularly updated.

  We propose to add the following information in Figure 2: networks that are updated in near-real-time, networks that are updated regularly, networks that are updated irregularly, and networks that are no longer being updated.

- Figure 2: The figure clearly shows the data availability by network. However, many networks are shown as active although no data has been provided within the last 5 or more years. As described not all networks have the capability to provide near-real time data, however I would suggest to highlight networks that are regularly providing data, even if that is in monthly or annual intervals.

  There are several networks that do not provide data regularly but of which we know that they are still recording data. For example, just after publication of this manuscript, the UMSUOL network provided their first data update in 8 years. We therefore propose to stratify the categories shown in networks that are updated in near-real-time, networks that are updated regularly (up till once per year), networks that are being updated irregularly, and networks that are no longer actively producing data (see revised figure below).

  1).

- Line 139ff: Perspectively, further data access options, such as an API would ease access to the data.

  This is a good suggestion which we will consider for future implementations.

[Figure]

Figure 1: Overview of all available networks, the individual time span of the data availability within the ISMN, their operational status and their update category (Status March 2021).

- Line 162: Please, specify what 'misuse' includes. Is this mostly the lack of referencing ISMN or are there other misuse cases (e.g. commercial use of the data)?

  This includes both a redistribution of the data and stealing of ground equipment. We propose to rephrase this into: *"...is required to prevent illegal redistribution of the data or theft of ground equipment, and to track ..."*.

- Figure 7: This is more a positioning comments as Fig 7 seems to belong to section 3.4 not 4.1

  We will make sure that in the final typesetting this figure is closely linked to Section 3.4

- Line 325: Reference missing (?)

  We will add the missing reference: "Bayat, B., Camacho, F., Nickeson, J., Cosh, M., Bolten, J., Vereecken, H., and Montzka, C.: Toward operational validation systems for globalsatellite-based terrestrial essential climate variables, International Journal of Applied Earth Observation and Geoinformation, 95, 102 240,https://doi.org/10.1016/j.jag.2020.102240, 2021."

- Section 5.1.4: The inclusion of citizen science data is shown by the GROW dataset where soil moisture sensors have been made available to citizens. Would this also be extended to categorical citizen science data from initiatives such as CrowdWater (https://crowdwater.ch) where soil moisture states are collected as categories instead of volumetric soil moisture values?

  We are aware of this interesting initiative and in the past even considered including such data in the ISMN. However, as the nature is very different from the other data and harmonising them with the rest is very difficult, we decided not to.

- Line 543 / Section 5.2.2: Long-term data series is not necessarily only linked to citizen science projects. I understand that as few standards as possible are defined in order to provide data to ISMN, however, in view of long-term operation, is there a minimum limit of data before inclusion into ISMN is considered? This does not necessarily concern citizen science projects alone but also short (2-5 year) soil moisture observations limited to project durations.

  Continuity is certainly also an issue for many conventional networks, but this is usually threatened by funding cuts rather than by discontinued engagement of the researchers. It is the long-term engagement of the citizens we refer to in this paragraph. In fact, we do have a policy in place requiring a minimum time series length of one year to be included in the ISMN. This allows to catch at least one full year with all seasonal influences (e.g., frozen soil, vegetation growth, rain season, dry season, etc.).

Minor comments:

- Line 81: Please correct to '... Data sets from ...'

  We will change this as suggested.

- Line 221: Please correct to '... absolute quality indicator ...'

  We will change this as suggested.

- Line 302: Please correct to '... downscale ...'

  We will change this as suggested.

- Line 595: Please correct to '... is required to ...'

  We will change this as suggested.

- Table A1: The TAHMO network goes beyond the Sahel Zone, maybe just drop the specification or change to (Sub-Saharan Africa)

  We will change this as suggested.

Citation: https://doi.org/10.5194/hess-2021-2-RC1

---

## Author Comment (AC3)

**1 Reply to Mirko Mälicke, 03 May 2021**

The authors present the International Soil Moisture Network (ISMN), a global database and collection of in situ soil moisture measurements. Beyond data storage, general issues about representativeness, data harmonization, and data quality are raised and answered. Finally, the extensive description of the ISMN state is accompanied by a review of scientific studies that used ISMN.

I think the manuscript is overall of high quality, great importance, clear language and I have fewer minor comments than the manuscript has authors.

With Best regards,
Mirko Mälicke

Dear Mirko Mälicke, thank you very much for your positive assessment of the manuscript and the valuable suggestions to further improve it. Our suggestions to improve the manuscript are given in blue font.

**My main comments are:**

– Section 4.2.1 This is the first section reviewing the usage of ISMN in scientific publications. The authors present a huge amount of studies, that use ISMN data to evaluate satellite products. The whole section is a collection of references and very brief descriptions, which satellite products were involved. I am wondering if there is any general conclusion that can be drawn from this section. Are there studies that found flaws in satellite products, which would not have been found without ISMN data? Did ISMN data and especially the fact that data from various networks is available, foster or even enable the development of novel satellite-based products, that would not have been possible?

How often is more than one network involved? Or to put it the other way around: Is the multitude of available networks driving satellite product evaluation or is ISMN a collection of isolated data that is used for evaluation anyway? There are so many great studies mentioned in this section, that it would be a waste to just list them without gaining some new insights.

These are all very good points, although it is be impossible to provide a conclusive answer to all of these questions. However, we suggest to add a subsection 4.7 at the end of chapter 4 where we discuss more explicitly the added value of the ISMN over the use of individual networks or over in situ soil moisture data obtained from elsewhere.

Approximately one third (34%) of the studies making use of ISMN used multiple networks, the choice of which depended on the scope of the study, the geographical region and period of interest, and the year the study was performed (with more and more data having become available over time). Examples of such studies using multiple networks are:

– Brocca et al. (2011) (6 ISMN networks used: IRPI, Campania, Calabria, UMSUOL, REMEDHUS, SMOSMANIA)
– Albergel et al. (2012) (3 ISMN networks used: REMEDHUS, UDC-SMOS, UMSUOL)
– Albergel et al. (2013) (5 ISMN networks used: SCAN, SMOSMANIA, REMEDHUS, MAQU, OzNet)
– Pellarin et al. (2013) (2 ISMN networks used: AMMA, USCRN)
– de Jeu et al. (2014) (10 ISMN networks used: AMMA, ARM, HYDROL-NET_PERUGIA, MAQU, MOL-RAO, OZNET, REMEDHUS, SCAN, SMOSMANIA, SNOTEL)
– Parinussa et al. (2014) (4 ISMN networks used: AMMA, REMEDHUS, SMOSMANIA, IIT Kanpur)
– Brocca et al. (2014) (4 ISMN networks used: SASMAS, SMOSMANIA, REMEDHUS, ICN)
– Paulik et al. (2014) (23 ISMN networks used: AMMA, ARM, CALABRIA, CAMPANIA, COSMOS, FLUXNET-AMERIFLUX, FMI, HYDROL-NET PERUGIA, ICN, IIT KANPUR, MAQU, MOL-RAO, OZNET, REMEDHUS, SASMAS, SCAN, SMOSMANIA, SNOTEL, SWEX POLAND, UDC SMOS, UMBRIA, UMSUOL, USDA-ARS)

45    – Balsamo et al. (2015) (6 ISMN networks used: AMMA, OzNet, SMOSMANIA, REMEDHUS, SCAN, SNOTEL)

– Brocca et al. (2015) (REMEDHUS used from ISMN but also other networks from different providers)

– Kim et al. (2015a) (2 ISMN networks used: COSMOS, USCRN)

– Kim et al. (2015b) (8 ISMN networks: COSMOS, FMI, PBO_H2O, REMEDHUS, SCAN, SOILSCAPE, TERENO, USCRN)

50    – Dirmeyer et al. (2016) (9 ISMN networks used: ARM, AWDN, COSMOS, PBO_H20, SCAN, SNOTEL, SOILSCAPE, USCRN, USDA ARS)

– Wang et al. (2016) (6 ISMN networks used: HOBE, MAQU, REMEDHUS, SASMAS, SMOSMANIA, TP (China))

– Scholze et al. (2016) (7 ISMN networks in US used)

– Al-Yaari et al. (2016) (4 ISMN networks used: AMMA, OZNET, REMEDHUS, SCAN)

55    – Enenkel et al. (2016) (4 ISMN networks used: COSMOS, DAHRA, REMEDHUS, SMOSMANIA)

– Kerr et al. (2016) (8 ISMN networks used: AMMA, ARM, DAHRA, OZNET, PBO-H2O, SCAN, SNOTEL, USCRN)

– Wu et al. (2016) (4 ISMN networks used: SCAN, COSMOS, USCRN, SNOTEL)

– Parinussa et al. (2016) (all available ISMN networks at that time used?)

60    – van der Schalie et al. (2016) (14 ISMN networks used: AMMA, ARM, COSMOS, FLUXNET-AMERIFLUX, OZNET, PBO H2O, REMEDHUS, SCAN, SMOSMANIA, SNOTEL, SOILSCAPE, SWEX POLAND, USCRN, VAS)

– Zhao et al. (2016) (6 ISMN networks used: CTP_SMTMN, ICN, OZNET, SCAN, SMOSMANIA, SNOTEL)

– Pan et al. (2017) (3 ISMN networks used: SCAN, SNOTEL, USCRN)

65    – Cui et al. (2017) (REMEDHUS from ISMN used and also one other network from another provider)

– Gruber et al. (2018) (2 ISMN networks used: SCAN, USCRN)

– Alemohammad et al. (2018) (10 ISMN networks used: FMI, iRON, REMEDHUS, RSMN, SCAN, SMOSMANIA, SNOTEL, SOILSCAPE, TERENO, USCRN)

– Eroglu et al. (2019) (2 ISMN networks used: OzNet, SCAN)

70    – Bauer-Marschallinger et al. (2019) (2 ISMN networks used: COSMOS, UMBRIA)

– Senyurek et al. (2020) (3 ISMN networks used: COSMOS, SCAN, USCRN)

– Link et al. (2020) (2 ISMN networks used: REMEDHUS, SCAN)

– Beck et al. (2020) (27 ISMN networks used: ARM, BIEBRZA, BNZ-LTER, COSMOS, CTP_SMTMN, DAHRA, FMI, FR_Aqui, HOBE, HYDROL-NET, iRON, LAB-net, MySMNet, ORACLE, OZNET, REMEDHUS, RISMA, RSMN, SCAN, SMOSMANIA, SNOTEL, SOILSCAPE, SWEX, TERENO, UDC, USCRN, WSMN)

75    – Liu et al. (2020) (3 ISMN networks used: REMEDHUS, NAQU, OZNNET)

– Kovačević et al. (2020) (3 ISMN networks used: PBO_H2O, SCAN, USCRN)

– Hagan et al. (2020) (20 ISMN networks used: AMMA-CATCH, ARM, CARBOAFRICA, COSMOS, CTP-SMTMN, DAHRA, FR-Aqui, HOBE, LAB-net, MAQU, OZNET, REMEDHUS, RSMN, SASMAS, SCAN, SMOSMANIA, SNOTEL, UMBRIA, UMSUOL, USCRN)

80

Although it is impossible to quantify precisely to what degree the ISMN has contributed to product improvements, there are certainly studies that discovered flaws in satellite products that would not have been detected without the use of the ISMN. The main reason is that the ISMN provides data from a large number of networks, thus giving insights in product skill in a large variety of climate zones, land cover types and so on, which no single network could have provided.

85 Secondly, the data harmonisation and unified quality control minimise the chance that differences in skill potentially observed between locations with different environmental characteristics is an artefact of a different treatment of the data. Examples of studies that not only identified flaws but really improved models/products etc. using data from the ISMN include:

- Beck et al. (2020), who calibrated 7 relevant parameters of the HBV model using in situ soil moisture measurements
90 from 177 independent sensors from the ISMN;

- Pan et al. (2017), who used ISMN soil moisture data to train and validate a new Artificial Neural Network based soil moisture product;

- Gruber et al. (2018), who assimilated spatially sparse ISMN soil moisture into a Continuous Model Domain: Gruber et al. (2018) over the Continental US to obtain improved model performance;

95 - Eroglu et al. (2019), who trained a neural network with ISMN data to derive high spatio-temporal resolution CYGNSS soil moisture estimates;

- Xu et al. (2018), who fused ISMN and GNSS-R data to obtain a superior soil moisture product over western Continental U.S.;

- Kang et al. (2019), who calibrated the SMOS V620 soil moisture retrieval algorithm for local site conditions in the
100 humid tropical regions of Malaysia, an area where the original product shows poor skill.

Even if it is likely that most products could have been developed without the use of the ISMN, in many cases the ISMN has made the development and evaluation more robust, both because of the large number of readily accessible data and because of the large networks and datasets that can be accessed only through the ISMN.

- Section 5.1.2 (P. 25): The authors describe challenges concerning the spatial representativeness of ISMN data and the
105 resulting issues for validating (spatially) coarser soil moisture data. From my point of view, similar issues can result from temporal inconsistencies. While the ISMN has made a lot of effort to harmonize the temporal resolution of ISMN data, this does not yet solve all representativeness issues. The authors should consider adding a short paragraph on temporal representativeness issues here, as well.

110 We agree that temporal representativeness issues may exist but that due to the standard sampling of 1 hour they have a smaller impact than spatial ones. We believe that a dedicated subsection on this issue would be too much, so instead we propose to add an additional paragraph at the end of the section:

"Also temporal representativeness issues may exist, but due to the hourly sampling of most datasets the ISMN usually have a denser sampling than most remote sensing or model data sets. Thus, for most applications the ISMN can be
115 downsampled to the process or observation timescale of interest. However, some of the older, manually sampled datasets have sampling intervals of about two weeks, and thus may miss many higher frequency wet or dry spells. On a similar note, datasets with a daily sampling or averaging (e.g., Cosmic ray or GPS reflectometry observations) may miss rainfall peaks and are unsuitable to study sub-daily variability."

- As a general comment, I would like to point out, that the first almost 20 pages of the manuscript are summarizing and
120 discussing the data in ISMN as well as ISMN itself. While I do think that all of the presented figures and sections are important and helpful, the more classic review of scientific studies using ISMN is noticeably shorter. I think in some parts, like section 4.2.1, the authors might consider extending section 4. Some more insights on how the reviewed studies are fostered or are even made possible in the first place, due to ISMN could be helpful. ISMN is not just a collection of data files, it's a reference database, that has the power to set standards and produce unified quality controls and metadata
125 information for soil moisture measurements. I would love to see a section elaborating how ISMN might already do this today, or what a possible path might look like for the future. Reviewing all the work that is based on or involving ISMN, the authors might see a pattern here.

This comment is related to the one above. We suggest to add an extra subsection to Chapter 4 on the added value of the ISMN over individual networks (See above). regarding setting standards, we propose the add the following sentence to section 4.2.2.: "The combination of ISMN, QA4SM, and enhanced quality control protocols and selection procedures to establish a set of fiducial reference measurements, is expected to become the standard for satellite soil moisture validation in the next years (Bayat et al., 2021).

**These are my minor comments in order of appearance:**

– P.2 L.10: Which proportion of the total datasets are still updated? A number could help the reader to set *"many"* into a context. The number is named on P. 4 L. 75 and the authors could consider moving this up. From my point of view, having roughly 70% of the networks still active is a feature and more than I would have expected.

We agree and mention this number also in the abstract.

– P.3 L. 36: At this point I was already so excited about the *advanced quality control methods*, that the authors could consider naming at least one of them as an example, already.

We appreciate your excitement about the advanced quality control methods but for the sake of the text flow and redundancy we believe it is better not to further elaborate on it at this point. Instead, we refer to the original paper describing these methods and to the section in the current paper describing the updates.

– P.4 L. 50 - 51: I personally like the idea of involving citizen scientists in the collection of soil moisture data and have already involved students (who are not 'citizens' in this case) in some short measurement campaigns. I personally doubt, that citizen scientists can really contribute data that hold metadata and quality control requirements of most scientific applications. Therefore, from a personal point of view, I would like to see at least one reference of a successful application here.

We agree that this is very challenging and certainly has caveats. We elaborate on this in Section **??**. We will add this reference.

– P. 4 L. 69: The authors might consider adding some numbers about the per-continent distribution of networks here, to give the reader a quick overview.

We added these numbers to main document.

– P. 4 L. 72: I am wondering if it would be feasible to add a graph about the landscape type distribution, here.

We were unsure what is meant with *landscape type*, but propose to include a plot showing the distribution per Köppen-Geiger climate class (Figure 1).

[Figure]

**Figure 1.** Distribution of ISMN-sites per Koeppen-Geiger climate classification, only categories with >20 sites are considered (Status June 2021).

– Figure 1 (P. 5) Due to the station density, I suggest putting a sub-figure of the USA into the map. Right now, I can only see that there are a lot of different stations of different colors. Another approach could be to rather indicate the geometric centroids of the networks in the USA and keep the stations in a transparent grey

This is a good point. We propose to plot the networks as in Figure 2. We believe that this is a better solution than plotting the centroids, as the location of the centroids is sometimes not representative for the actual location of the stations, particularly in large countries or for networks spanning multiple continents.

2).

[Figure]

**Figure 2.** Locations of ISMN networks and sites plotted with the ISMN package described in the *Code and data availability* section [Status June 2021].

– Figure 2 (P. 6) I am a little bit confused by figure 2. I can see many networks that are marked as active but did not contribute any data for years. How is this possible? What exactly is an active network?

165    We appreciate the confusion caused by several networks that are still active but only irregularly updated (with a potential delay of up till several years) in the ISMN. We propose to update the graphic and stratify active networks according to the frequency of updating, i.e., in NRT, regularly (up till once a year), and irregularly updated (Figure 3).

3).

[Figure]

**Figure 3.** Overview of all available networks, the individual time span of the data availability within the ISMN, their operational status and their update category (Status June 2021).

– Table 1 (P. 7) I think this table could really be enhanced by adding another column containing the total count of sensors / timeseries / stations, whatever technically appropriate, of the respective variable. I.e. for soil suction, it will be of importance for many readers, how often soil moisture data is accompanied by suction data.

This is a good point. We propose to expand the table as in 1

Table 1: Overview of all available temporally dynamic variables stored in the ISMN database. *Note that for precipitation and air temperature the measurement height above the ground surface is indicated.

| variable name | abbreviation | units | measurement depth* [m] | variable with depth? | number of time series (stations) |
|---|---|---|---|---|---|
| soil moisture | sm | $m^3 m^{-3}$ | 0.00 - 2.10 | Y | 10610 (2822) |
| soil suction | su | kPa | 0.04 - 0.75 | Y | 73 (18) |
| soil temperature | ts | °C | 0.00 - 2.03 | Y | 8113 (1629) |
| air temperature | ta | °C | 2.00 - 12.00 | Y | 1292 (1234) |
| surface temperature | tsf | °C | 0.00 - 0.00 | N | 126 (126) |
| precipitation | p | mm | 0.00 - 2.00 | N | 759 (700) |
| snow depth | sd | mm | 0.00 | N | 562 (555) |
| snow water equivalent | sweq | mm | 0.00 | N | 507 (427) |

– P.9 L. 151: The authors state, that the data is formatted according to CEOP or a slightly modified version. Does this imply that the data can come in different formats, or is it always the modified format (which was derived from CEOP)?

Indeed, the formatting can be chosen by the user. We will rephrase the sentence to make this clearer.

P. 10 L. 159 - 160: I was just wondering if the authors think, that this user feedback (on quality issues) might also be a valuable public resource (similar to Github issues)? Having a 'conversation' about quality issues publicly, might help other users to handle these issues for their specific application.

This is a good idea. Several years ago, we already started communicating dataset issues via the forum, but this has never really broadly adopted by the users. Github is currently used for the ISMN reader and quality control package (FlagIt) but is more on the developer side than for reporting general issues (e.g., how to download, how to access data, how to participate, etc.) or discuss data quality (e.g., more detailed timeseries issues, specific station information, etc.). Also, we operate a Twitter channel which is likely to reach out more easily to users. We are currently thinking of ways how we can coordinate the various outreach channels to disseminate information more efficiently and with maximum outreach.

– P. 10 L. 161 - 162: How exactly can a registered download option prevent data misuse? Isn't it the distributed license that regulates the terms and conditions of how the data may be used? Comparable to this manuscript itself, which can also be retrieved without registration and it's the CC BY 4.0 that prevents me from misusing it.

Theoretically, a license agreement should cover this but in practice this can hardly be avoided as it is difficult to prove how the data have been used. However, the registration mainly serves as a filter to gaining access to the exact station coordinates, as there have been reported cases of stolen equipment from observation sites.

– P. 11 L. 193 - 195: I was a bit confused by this paragraph. Does it mean that an additional flag for the other variables soil temperature, air temperature, and precipitation was added, or that a spurious observation in one of these variables will also flag the soil moisture measurement as spurious?

We agree that this formulation is confusing. We therefore rephrased it to: "In case more than one soil temperature, air temperature, or precipitation sensor is available at a site, a flag is raised for the soil moisture measurement if the conditions of flags D01, D02, and D04, respectively is met at least for one of these sensors."

– P. 15 L. 239: What is ubRSMD?

unbiased Root-Mean-Squared Difference (ubRMSD). We spelled out this acronym in the main text.

- P. 16 L. 248: A short clause describing what the "triple collocation approach" is would help me to understand this section even better.

  We propose to rephrase this section to clarify it: " Gruber et al. (2013) adopted a triple collocation approach (TCA) to characterise the representativeness errors of ISMN data for coarse-scale (∼25 km) use. TCA is a statistical analysis using a combination of three datasets with independent random errors to estimates the error variance in each of these datasets. Here, we apply the TCA to estimate the representativeness errors of the ISMN data of all networks with sufficient sampling in the period 2001-2019 for application at the coarse scale. ESA CCI SM passive soil moisture (v5.2, Gruber et al. (2019b),Dorigo et al. (2017)) and top-layer ERA5 volumetric soil water content (Hersbach et al., 2020) are used to complement the triplets.

- P 20 L. 325: I guess the question mark should not be there.

  resolved

- Section 4.2.2 (P.20 - 21) This section gives an overview of operational services relying on ISMN data. Unfortunately, the authors only refer to the 'ISMN data'. I think it could be more helpful for readers, that might happen to be users of the services as well if the specific networks involved for each of these services are listed. Otherwise, I am under the impression, that all data is involved all the time.

  Unfortunately, it is impossible for us to trace back which networks were exactly used. The same applies to many other peer-reviewed validation studies. This has been one of our main motivations to initiate QA4SM which aims for full transparency of the entire validation chain, including network and station selection. An example is given in https://doi.org/10.5281/zenodo.4736927 and in https://doi.org/10.5281/zenodo.4120205 for C3S and ESA CCI Soil Moisture validation, respectively. We propose to add these examples to the main document.

- Section 5.1.3 For me it was not clear from this section, if ISMN nowadays already includes a considerable amount of low-cost sensor networks and if the raised issues could be addressed using ISMN. The authors should consider clarifying this.

  We propose to add the following sentence: "The ISMN has integrated low-cost sensor measurements from the GROW observatory (see next section), but a number of practical challenges arise when integrating such data.

- Section 5.1.4 This is connected to the point above: Is GROW part of ISMN? I guess it is, as it is listed in the appendix. Further, this section points at some challenges connected to citizen science, but it remains kind of open whether citizen science data should be contributed or not. The authors should make this more clear. A scientific user of ISMN should be aware of the context of data collection and I am not sure if the metadata provided is suitable enough to transport this important information if it is even contained.

  We propose to add the following sentence to clarify this: "An outstanding example of a citizen science project focusing on soil moisture is the GROW Observatory (growobservatory.org), from which datasets panning at least a year of data have been added to the ISMN."

  As crowd-sourced data can have completely different characteristics and properties, it is not possible to give a simple answer whether such measurements should be included or not. We addressed the most important points which should be considered before inclusion, such as: thorough evaluation of the (low-cost) sensors used (lines 522-525), training citizens on installation and maintenance of the sensors (lines 538-540), visual inspection of the timeseries (when possible - lines 552-553), presence of few/several sensors within satellite footprint in order to reduce uncertainties (lines 550-552). If all the above mentioned issues have been carefully addressed (i.e. reflecting a well-organized citizen science project - lines 554-555), then crowd-sourced data can be included into the ISMN. As crowd-sourced data can have completely differerent characteristics and properties, it is not possible to give a simple answer wether such measurements should be included or not. We addressed the most important points which should be considered before inclusion, such as: thorough evaluation of the (low-cost) sensors used (lines 522-525), training citizens on installation and maintenance of the sensors (lines 538-540), visual inspection of the timeseries (when possible - lines 552-553), presence of few/several sensors

within satellite footprint in order to reduce uncertainties (lines 550-552). If all the above mentioned issues have been carefully addressed (i.e. reflecting a well-organized citizen science project - lines 554-555), then crowd-sourced data can be included into the ISMN.

**Finally, I would like to make two personal comments, that are not part of the review in a strict sense:**

- Concerning 5.2.3: Working on a web-based data portal for a couple of years myself, I would suggest adding a new section to the data overview Bubbles on the map, that contains a suggestion on how to cite a specific dataset. The same could be implemented for networks and on download. I think this will already significantly increase the visibility and citations, as they are usually not missed out on bad will.

  Yes this is indeed a great idea. Why did we not think of that? We would need to adjust the display query as well as the bubble itself but it should be manageable. To your information we have just recently added a separate database table for citations. With this table we want to integrate a file per download containing a list of citation information for each individual network.

- Secondly, I personally don't want to finish my comment, without mentioning ESSD. The authors have described the state of the ISMN database in great detail and will convince every reader why the database and the data itself are unique and a real asset for environmental science. While this manuscript is labeled as a review paper, a considerable amount of it is also halfway a dataset description paper. From my personal point of view, it could be really worth it to prepare an ESSD publication for ISMN itself. That can help to further increase the visibility of ISMN. At the end of the day, I found myself a couple of times hitting 'soil moisture' into the search bar of ESSD, and not finding ISMN is a pity.

  We have indeed thought of submitting a manuscript to ESSD, but unfortunately the ISMN does not qualify for this journal. Since the ISMN requires a (free) registration, as requested by our data providers to prevent access to sensitive information (as mentioned before), this is not considered completely open access.

With Best regards,
Mirko Mälicke

**References**

[revised manuscript text omitted]